



# Temporal and spatial analysis of ozone concentrations in Europe based on time scale decomposition and a multi-clustering approach

Eirini Boleti[1,2], Christoph Hueglin[1], Stuart K. Grange[1,4], André S. H. Prévôt[3], and Satoshi Takahama[2]

[1]Empa, Swiss Federal Laboratories for Materials Science and Technology, Überlandstrasse 129, Dübendorf, Switzerland
[2]EPFL,École Polytechnique Fédérale de Lausanne, Route Cantonale, 1015 Lausanne, Switzerland
[3]PSI, Paul Scherrer Institute, 5232 Villigen, Switzerland
[4]Wolfson Atmospheric Chemistry Laboratories, University of York, York, YO10 5DD, UK

*Correspondence to:* Christoph Hueglin (Christoph.Hueglin@empa.ch)

**Abstract.**

Air quality measures that were implemented in Europe in the 1990s resulted in reductions of ozone precursors concentrations. In this study, the effect of these reductions on ozone is investigated by analyzing surface measurements of ozone for the time period between 2000 and 2015. Using a non-parametric time scale decomposition methodology, the long-term, seasonal and

short-term variation of ozone observations were extracted. A clustering algorithm was applied to the different time scale variations, leading to a classification of sites across Europe based on the temporal characteristics of ozone. The clustering based on the long-term variation resulted in a site type classification, while a regional classification was obtained based on the seasonal and short-term variations. Long-term trends of de-seasonalized mean and meteo-adjusted peak ozone concentrations were calculated across large parts of Europe for the time period 2000-2015. A multi-dimensional scheme was used for a detailed

trend analysis, based on the identified clusters, which reflect precursor emissions and meteorological influence either on the inter-annual or the short-term time scale. Decreasing mean ozone concentrations at rural sites and increasing or stabilizing at urban sites were observed. At the same time downward trends for peak ozone concentrations were detected for all site types. The effect of hemispheric transport of ozone can be seen either in regions affected by synoptic patterns in the northern Atlantic or at sites located at remote high altitude locations. In addition, a reduction of the amplitude in the seasonal cycle of ozone was

observed, and a shift in the occurrence of the seasonal maximum towards earlier time of the year. Finally, a reduced sensitivity of ozone to temperature was identified. It was concluded that long-term trends of mean and peak ozone concentrations are mostly controlled by precursors emissions changes, while seasonal cycle trends and changes in the sensitivity of ozone to temperature are driven by regional climatic conditions.

## 1    Introduction

Tropospheric ozone ($O_3$), together with particulate matter and nitrogen dioxide ($NO_2$), is one of the most troublesome air pollutants in Europe (EEA, 2016). 17,000 premature deaths every year are attributed to excess $O_3$ exposure, without any sign of reduction in number of fatalities (EEA, 2016). In terms of impact on ecosystems, elevated concentrations of tropospheric $O_3$ are responsible for damaging agricultural production and forests mainly by reducing their growth rate. In addition, tropospheric





$O_3$ acts as a greenhouse gas with an estimated globally averaged radiative forcing of $0.4\pm0.2\,W/m^2$ (IPCC, 2013). In the 1990s emission control measures on $O_3$ precursors, namely nitrogen oxides ($NO_x$=NO+NO$_2$) and volatile organic compounds (VOCs), were implemented in order to regulate air pollution. As a result, concentrations of $NO_x$ and VOCs have significantly declined in Europe (EEA, 2017; Colette et al., 2011; Guerreiro et al., 2014; Henschel et al., 2015). Especially, $NO_x$ emissions

declined in Europe by 48% between 1990 and 2015 (EEA, 2017).

Surprisingly, $O_3$ concentrations have not decreased as was expected (Oltmans et al., 2013; Colette et al., 2018). Mean $O_3$ concentrations have either remained stable or even increased in rural, background areas from 1990s and until mid-2000s in many European countries (Boleti et al., 2018a; Munir et al., 2013; Paoletti et al., 2014; Querol et al., 2016; Anttila and Tuovinen, 2009, e.g. ). At urban sites an increase of mean $O_3$ has been observed; in some cases, an increase has been found at

both rural and urban sites with larger upward trends observed at urban compared to the rural sites (Paoletti et al., 2014; Querol et al., 2016; Anttila and Tuovinen, 2009). However, a change in the trend has been observed after mid-2000s, when mean $O_3$ concentrations have started to decline (Boleti et al., 2018a; Munir et al., 2013). On the other hand, maximum $O_3$ concentrations decreased continuously from the 1990s until present (Paoletti et al., 2014), except for the traffic loaded environments (Boleti et al., 2018b). Downward trends of different metrics for peak $O_3$ have been found at many sites across Europe (Fleming et al.,

2018). However, the high year to year variability of $O_3$ tends to mask the long-term changes leading to a large fraction of sites with non-significant trends . Several studies based on either observations or climate models have shown that anthropogenic emissions can affect $O_3$ concentrations across continents (Dentener et al., 2010; Wild and Akimoto, 2001; Lin et al., 2017). The increase of background $O_3$ in Europe has been associated with increasing stratospheric $O_3$ contribution (Ordóñez et al., 2007), as well as increased hemispheric transport of $O_3$ and its precursors.

A shift in the seasonal cycle of $O_3$ has been observed in northern mid-latitudes, i.e. the peak concentrations are now observed earlier in the year compared to previous decades with a rate of 3-6 days/decade. (Parrish et al., 2013). This shift is attributed to increasing emissions of $O_3$ precursors in developing countries, that led to an equatorward redistribution of precursors in the previous decades (Zhang et al., 2016). Negative trends of the 95th percentile of $O_3$ and positive trends for the 5th percentile have been detected across Europe (Yan et al., 2018). Simultaneous decrease of maximum concentrations in summer and increase

in winter indicate a decrease of amplitude in the seasonal variation of $O_3$, probably as a result of the regulations in the 1990s (Simon et al., 2015).

$O_3$ variations are largely governed by climate and weather variability (Yan et al., 2018). Especially temperature influences $O_3$ concentrations in the troposphere, mainly by increasing the rates of several chemical reactions, and by increasing emissions of biogenic VOCs with increasing temperature (Sillman and Samson, 1995). Thermal decomposition of peroxyacyl nitrates

(PANs) at high air temperature conditions results in elevated $O_3$ concentrations (Dawson et al., 2007). Indeed, extreme $O_3$ concentrations in central Europe are mainly associated with high temperatures (Otero et al., 2016). However, there are indications that the relationship of $O_3$ to temperature has changed in the last 20 years. For instance, in the U.S., $O_3$ climate penalty – defined as the slope of the $O_3$ versus temperature relationship – dropped from 3.2 ppbv/°C before 2002 to 2.2 ppbv/°C after 2002 as a result of $NO_x$ emission reductions (Bloomer et al., 2009).

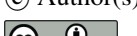



At different locations, $O_3$ may show a different temporal evolution due to a variety of factors, such as local pollution, topography, influence of nearby sources or even trans-boundary transport of $O_3$ and its precursors. In addition, meteorological conditions can vary amongst different locations within large regions such as Europe, affecting $O_3$ concentrations in various ways. $O_3$ trend studies in the past have tried to tackle this issue, mainly by using clustering techniques to categorize European

measurement sites based on different $O_3$ metrics (e.g. Henne et al. (2010)). For instance, a site type classification representing $O_3$ regimes between 2007 and 2010 was obtained by Lyapina et al. (2016) using mean seasonal and diurnal variations. In addition, a geographical categorization reflecting the synoptic meteorological influence on $O_3$ variation between 1998 and 2012 was obtained by Carro-Calvo et al. (2017). To tackle low spatial representation of urban and rural sites across large domains, i.e. mid-latitude North America, western Europe and East Asia, Chang et al. (2017) obtained a latitude dependent site

classification with lower concentrations in western and northern Europe and higher concentrations in southern Europe. These studies indicate that the selected metric used to characterize $O_3$ in clustering leads to site classifications that represent different aspects of $O_3$ variability.

In the current study, a multidimensional clustering method that captures several influencing factors for the long-term trend of $O_3$ is presented. The temporal and spatial evolution of $O_3$ concentrations between 2000 and 2015 is studied using data provided

by the European Environmental Agency (EEA). Mean $O_3$ concentrations are decomposed into the underlying frequencies based on a non-parametric time scale decomposition method to obtain the long-term (LT), seasonal (S) and short-term (W) variations. The multidimensional clustering approach is applied to the distinct frequency signals LT(t) and S(t) extracted from the observations.

In addition, long-term trends of de-seasonalized daily mean $O_3$ and meteo-adjusted peak $O_3$ concentrations are calculated.

Through de-seasonalization and meteo-adjustment, a significant fraction of the meteorologically driven variability of $O_3$ is excluded from the observations, and uncertainty in the trend estimation is reduced by a large factor. Intersections of site groups, i.e. LT(t)-and S(t)-clusters, are employed to guide the study of $O_3$ long-term trends. Furthermore, changes in the amplitude and phase of the seasonal variability of $O_3$ are explored based on the S(t) signal obtained by the time scale decomposition methodology. Finally, long-term changes in the relationship between $O_3$ and temperature are estimated and discussed for the

different site environments and regions in Europe.

## 2 Data

Data for $O_3$ surface measurements are provided by the EEA (Air Quality e-Reporting) in an hourly resolution for the period between 2000 and 2015. In this study, only time series with a maximum of 15% of missing values, and a maximum of 120 consecutive days with missing values are used, leaving the study with 291 sites across the European domain (Fig. 1). The daily

mean and the daily maximum of the 8 hour running mean based on hourly mean concentrations (MDA8 $O_3$) are calculated following the definition by the European Union Directive of 2008 (European Parliament and Council of the European Union, 2008). For the representation of peak concentrations the following metrics are used: (a) MTDM, which is the mean of the ten





highest daily maximum $O_3$ concentrations during May and September based on hourly mean data and (b) 4-MDA8, the mean of the four highest MDA8 $O_3$ concentrations per year.

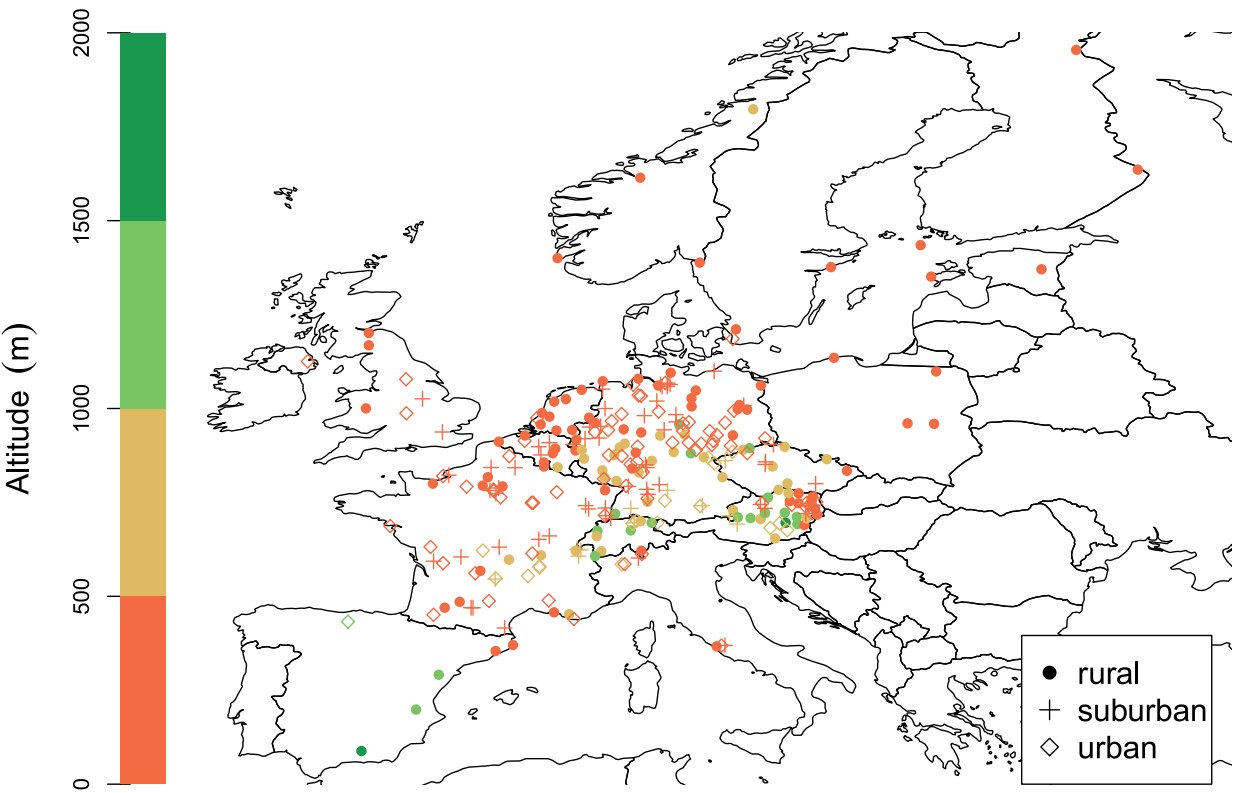

**Figure 1.** European map showing the location of the studied sites. Type of environment (symbols) and altitude (color bar) are indicated.

Meteorological variables are extracted from the ERA-Interim data-set on a 1 degree grid at the location (longitude-latitude-altitude) of each station and in 3-hourly intervals. The variables considered for the meteo-adjustment of the peak $O_3$ metrics are temperature (K), specific humidity (kg/kg$^{-1}$), surface pressure (hPa), boundary layer height (m), convective available potential energy (CAPE, J$\cdot$kg$^{-1}$), East-West surface stress (N$\cdot$s$\cdot$m$^2$) and North-South surface stress (N$\cdot$s$\cdot$m$^2$).

The present trend analysis focuses on (a) the de-seasonalized daily mean and MDA8 $O_3$ and (b) the meteo-adjusted MTDM and 4-MDA8 concentrations. The analysis of changes in the seasonal cycle of $O_3$ across Europe is based on the daily mean $O_3$ concentrations.





## 3 Methods

### 3.1 Time scale decomposition of daily mean and MDA8 $O_3$

Time scale decomposition refers to decomposition of the $O_3$ time series into the relevant underlying frequencies:

$$O_3(t) = LT(t) + S(t) + W(t) + E(t) \tag{1}$$

where $O_3(t)$ is the daily mean and MDA8 $O_3$ time series, $LT(t)$ the long-term variation, $S(t)$ the seasonal variation, $W(t)$ the short-term variation and $E(t)$ the remainder of the decomposition. Time scale decomposition in this study is performed with a non-parametric method, called the ensemble empirical mode decomposition (EEMD, Huang et al., 1998; Huang and Wu, 2008; Wu and Huang, 2009), which is considered a powerful method for decomposing $O_3$ time series (Boleti et al., 2018a). The method detects the hidden frequencies in the time series based merely on the data and yields the so-called intrinsic mode

functions (IMFs); each IMF represents one distinct frequency in the signal.

$$y(t) = \sum_{j=1}^{n} c_j(t) + LT(t) \tag{2}$$

where $y(t)$ is the input data, $c_j$ the different IMFs, $n$ the number of the IMFs and the remainder time series is the LT(t) of the input data. By adding together the IMFs with frequencies between around 40 days and 3 years we obtain the seasonal variation of $O_3$ ($S(t) = c_7 + ... + c_{10}$) and by adding the frequencies that are smaller than 40 days the short-term variation is acquired

($W(t) = c_1 + ... + c_6$).

### 3.2 Cluster analysis of $O_3$ variations

Cluster analysis is referred to pattern recognition in high dimensional data. The main idea is to represent $n$ objects by identifying $k$ groups based on levels of similarity. Objects in the same group must have the highest level of similarity while objects from different groups must have low level of similarity (Jain, 2010). The partitioning around medoids (PAM) clustering algorithm

is used in this study. It is based on k-means (MacQueen, 1967; Hartigan and Wong, 1979) which is a widely used clustering technique (Lyapina et al., 2016, e.g. ). PAM is more robust than k-means, because it minimizes the sum of dissimilarities instead of the sum of squared euclidean distances (R Development Core Team, 2017). It works as follows: First, a set of $n$ high dimensional objects (measurement sites) is clustered into a set of $k$ clusters. Initially, $k$ clusters are generated randomly and the empirical means $m_k$ of the euclidean distance between their data points are calculated. Then, each data point is assigned to

its nearest cluster center (centroid). Centroids are iteratively updated by taking the medoid of all data points assigned to their clusters. The squared error ($\varepsilon$) between the $m_k$ and the points in the cluster ($x_i$) is calculated as:

$$\varepsilon = \sum_{i=1}^{n} \|x_i - m_k\|^2 \tag{3}$$

Each centroid defines one of the clusters and each data point is assigned to its nearest centroid, and the iterative process is terminated when the $\epsilon$ is minimized.





For identification of the clusters the LT(t), S(t) and W(t) of the daily mean and MDA8 $O_3$ were used as input time series in the PAM algorithm. A sufficient number of clusters must be defined in order to capture dominant behaviors such that redundant information is avoided but at the same time not overlooking important characteristics. To identify the optimal number of clusters the k-means algorithm is iteratively executed for a range of k values (number of clusters) and the average sum of $\epsilon$ (SSE) is
calculated for each iteration, i.e. each k.

$$SSE = \sum_{i=1}^{n} \varepsilon^2 \tag{4}$$

The number of clusters with the largest reduction in $SSE$ is considered as the most representative. Eventually, the choice of the ideal number of clusters results from a combination of the $SSE$ approach and interpretability of the obtained clusters. In addition, a Silhouette width ($S_w$) analysis is performed to assess the goodness of the clustering (Rousseeuw, 1987).
More details about the number of clusters, the goodness of the clustering and the $S_w$ are provided in the supplementary material.

### 3.3  Daily mean and MDA8 $O_3$ long-term trend analysis

Meteorological adjustment is essential for calculation of robust $O_3$ long-term trends. Thus, daily mean and MDA8 $O_3$ observations are de-seasonalized by subtracting the S(t) obtained with the EEMD from the observations (Boleti et al., 2018a)

$$y_d(t) = y(t) - S(t) \tag{5}$$

where $y_d(t)$ the de-seasonalized time series and $y(t)$ the observations. Through de-seasonalization observations are adjusted for the effect of meteorology on the inter-annual time scale. Theil-Sen trends (Theil, 1950; Sen, 1968) are then calculated based on monthly mean de-seasonalized concentrations of the $y_d(t)$ for the period 2000-2015. The 95% confidence interval of the
trend is obtained by bootstrapping. The Theil-Sen trends were estimated using the *openair* library in R (R Development Core Team, 2017).

### 3.4  Peak $O_3$ concentrations long-term trend analysis

Trend analysis of peak $O_3$ metrics is performed for the MTDM and the 4-MDA8 $O_3$, based on a meteo-adjustment approach as in Boleti et al. (2018b). A different approach for meteorological adjustment was used for the peak $O_3$ than for the daily mean
and MDA8, because de-seasonalization is not meaningful for peak $O_3$ because peak ozone events are temporally localized. Thus, daily maximum and MDA8 $O_3$ observations were linked to the available meteorological variables through generalized additive models (GAMs, Hastie and Tibshirani, 1990; Wood, 2006) for the warm season (May-September). GAMs are instances of generalized linear models in which the model is specified as a sum of smooth functions of the covariates. A GAM can be described as:

$$O_3(t) = \alpha + \sum_{i=1}^{n} s_i(M_i(t)) + s_0(t) + \epsilon(t) \tag{6}$$





where $O_3(t)$ stands for the $O_3$ time series observations (daily maximum and MDA8), $\alpha$ is the intercept, $s_i$ are the smooth functions (thin plates splines) of the numeric meteorological variables $M_i$ and $n$ denotes the number of the numeric meteorological variables in the GAM. The temporal trend is represented through the smooth function $s_0(t)$, where $t$ is the time variable expressed by the Julian day. Finally, $\epsilon$ stands for the residuals of the model. For the GAMs, the following meteorological

variables were used based on the meteorological variable selection performed by Boleti et al. (2018b): the daily maximum temperature, daily mean specific humidity, daily mean surface pressure, daily maximum boundary layer height, morning mean convective available potential energy (CAPE), daily mean East-West surface stress and daily mean North-South surface stress, as well as the Julian day. The GAMs were estimated with the *mgcv* library in R (R Development Core Team, 2017).

The meteo-adjusted daily maximum and MDA8 $O_3$ concentrations were calculated similar to Barmpadimos et al. (2011) as:

$$O_{3_{metadj}}(t) = \alpha + s_0(t) + \epsilon(t) \tag{7}$$

where $\alpha$ is the intercept of the model, $s_0(t)$ the time variable as Julian day, and $\epsilon(t)$ the residuals. The meteo-adjusted MTDM and 4-MDA8 concentrations were estimated based on the meteo-adjusted values ($O_{3_{adj}}(t)$) on the same days as they were identified before the meteo-adjustment. Eventually, meteo-adjusted trends were calculated with the Theil-Sen trend estimator

applied on the $O_{3_{metadj}}(t)$.

## 3.5  $O_3$ seasonal cycle trend analysis

The S(t) signal extracted with the EEMD captures the meteorologically driven $O_3$ variation on yearly to multi-year time scales, and is more representative compared to parametric fitting approaches (Boleti et al., 2018a). Here, changes in the daily mean S(t) of $O_3$ throughout the studied period are identified as follows: the maximum and minimum $O_3$ value as well as the day when

the maximum $O_3$ occurred in each year are identified in the S(t), referred here as $S_{max}$, $S_{min}$ and $S_{DoM}$ respectively (Fig. 2). A Theil-Sen trend estimator for each of the $S_{max}(t)$, $S_{min}(t)$ and $S_{DoM}(t)$ is applied for each site cluster, representing the long-term temporal evolution of the amplitude and phase of S(t).

## 3.6  Relationship between $O_3$ and temperature

The relationship between $O_3$ and temperature is studied for the warm season May-September. A linear regression model

between daily maximum $O_3$ concentrations and daily maximum temperature is applied for each year throughout the studied period 2000-2015 as:

$$O_3(t)_i = \beta_{0i} + \beta_{1i} \cdot T(t)_i, i = 1, 2, ..n \tag{8}$$

where $O_3(t)$ is the time series of the daily maximum $O_3$, $T(t)$ the time series of the daily maximum temperature and $n$ is the number of years. $\beta_{0i}$ is the intercept and $\beta_{1i}(t)$ the parameter describing the linear effect of temperature on $O_3$. Then, a

linear model is applied to $\beta_{1i}(t)$ over all years for each site cluster to identify the long-term trend of the slope between $O_3$





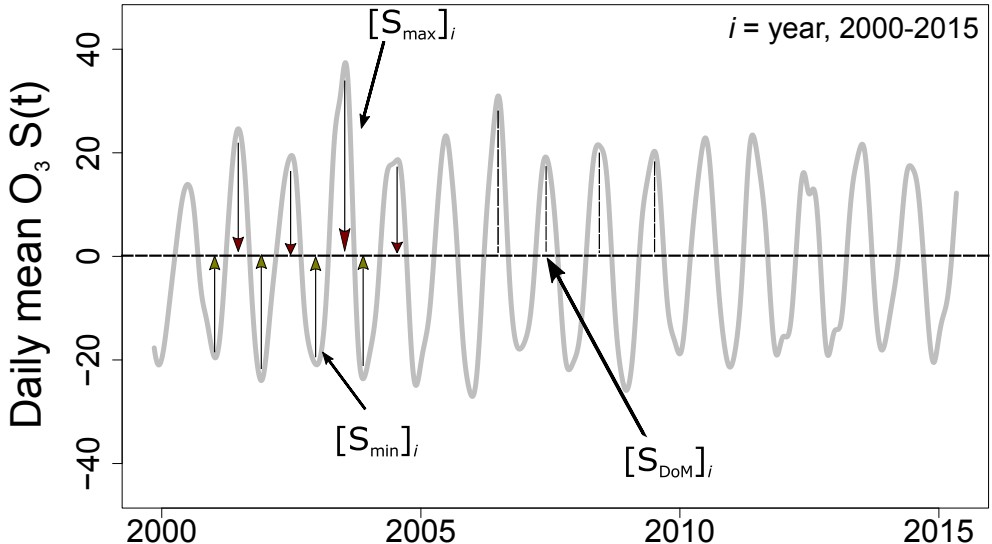

**Figure 2.** Schematic illustration for explaining the estimation of the ($S_{max}(t)$ and $S_{min}(t)$) and the annual day of maximum of the seasonal signal ($S_{DoM}(t)$) as calculated from the daily mean $O_3$ time series.

and temperature maximum values. In addition, a linear regression model is applied on the daily maximum $O_3$ concentrations against binned temperature ranges and in three consequent time periods (2000-2005, 2005-2010 and 2010-2015).

## 4 Results

### 4.1 Cluster analysis

5 Here, we present the results of the daily mean LT(t)- and S(t)-clustering; results for the W(t)-clustering and the cluster analysis based on the MDA8 are provided in the supplementary material. Application of the clustering algorithm to the LT(t) leads to a site type classification, which largely reflects the proximity to emission sources of $O_3$ precursors. S(t)- and W(t)-clustering leads to a regional site classification, which reflects the importance of the climate on the annual cycle of $O_3$. It is observed that a few sites have a negative $S_w$, which means that these sites are assigned to a certain cluster although they do not really fit into

10 any of the identified clusters (see supplementary material sections S2 an S5). Nevertheless, the sites with negative $S_w$ were not excluded from the data analysis as they do not have a noticeable influence on the results.

$O_3$ concentrations often increase with distance from emission sources of $NO_x$. Thus, LT(t)-clustering leads to identification of site groups with similar type of environment in terms of proximity to precursor emissions and mean $O_3$ concentrations,





which are indicative of multiannual changes in the $O_3$ time series (Boleti et al., 2018a). This measurement-based classification can be more informative than reported station types, since e.g. there are rural sites with nearby pollution sources or even sites with surroundings that might have changed dramatically with time. In the following section, clusters obtained from analysis of daily mean $O_3$ are presented and discussed in detail; the clusters derived from MDA8 $O_3$ are similar and presented in the

supplementary material.

      Cluster analysis of the long-term variation LT(t) resulted in four clusters that mainly differ in the daily mean $O_3$ concentration levels: Cluster 1 includes sites that are marked in the Air Quality e-Reporting data repository as being of rural site type and sites that are mostly located at higher altitudes (on average 800 to 1200 m). The sites in cluster 1 show the highest $O_3$ concentrations as illustrated in Fig. 3. The high mean $O_3$ concentrations indicate that the sites in cluster 1 are representing

background situations with minor influence of nearby emissions of man-made $O_3$ precursors. This cluster is therefore denoted as background cluster ("BAC"). Cluster 2 includes mostly rural sites, that are located at lower altitudes of around 300-600 m and is therefore labelled as rural cluster ("RUR"). The sites in cluster 3 are also located at low altitudes (around 100 to 300 m) and represent rural, suburban and urban site types in similar numbers. The sites in cluster 3 seem to be influenced by nearby man-made emissions of $O_3$ precursors such as $NO_x$, leading to lower mean $O_3$ concentrations compared to the sites in the

"RUR" cluster. Cluster 3 consists of moderately polluted sites and denoted as cluster MOD. Finally, cluster 4 consists mostly of urban and suburban sites showing the lowest daily mean $O_3$ concentrations likely due to the proximity to sources of $NO_x$ emissions and the resulting enhanced depletion of $O_3$ through reaction with NO. Consequently, cluster 4 is denoted as the highly polluted cluster ("HIG").

      The LT(t) signal as derived from the daily mean and MDA8 $O_3$ observations increases for "BAC" and "RUR" until around

beginning of 2000s and decreases afterwards. For the "MOD" and "HIG" clusters the same pattern was observed, but the decrease starts much later than in the rural sites, i.e. around end of 2000s. Especially in the "HIG" sites mostly a level-off is observed after 2010 rather than a decrease. Similar temporal evolution with inflection points in the LT(t) has been observed in the study by Boleti et al. (2018a) which was focused on trends of average $O_3$ concentrations in Switzerland.

      Clusters derived from the daily mean S(t) show a regional representation most likely due to the influence of the climatic

conditions an the annual cycle of $O_3$. The following five clusters were obtained from the daily mean S(t) (Fig. 4): (1) "CentralNorth" comprises northern and eastern part of Germany, Netherlands and some eastern sites in Czech Republic, Poland and Austria, (2) "CentralSouth" covers most part of Austria, Switzerland and some sites in the Southwest of Germany, (3) "West" incorporates the biggest part of France, Belgium and Spain, (4) "PoValley" includes the sites located in the Po Valley, an industrial region in Northern Italy. (5) "North" covers most of the UK and Scandinavia. The sites in "PoValley" display the

most pronounced S(t), mainly due to the Mediterranean weather conditions, e.g. high temperatures. At the same time high $NO_x$ and VOC emissions in this region leads to higher $O_3$ concentrations. The "North" cluster has the smallest seasonal variability, due to generally low $O_3$ concentrations, and lower temperature conditions in this region. Especially in the Scandinavian sites meteorological conditions are rather unfavorable for $O_3$ formation. Also, the regions included in the "North" cluster are influenced by cyclonic systems arriving in Europe through the North Atlantic ocean, that carry air pollutants into Europe (Stohl,

2002; Dentener et al., 2010). Thus, the influence of background $O_3$, i.e. $O_3$ inflow from northern America and eastern Asia,





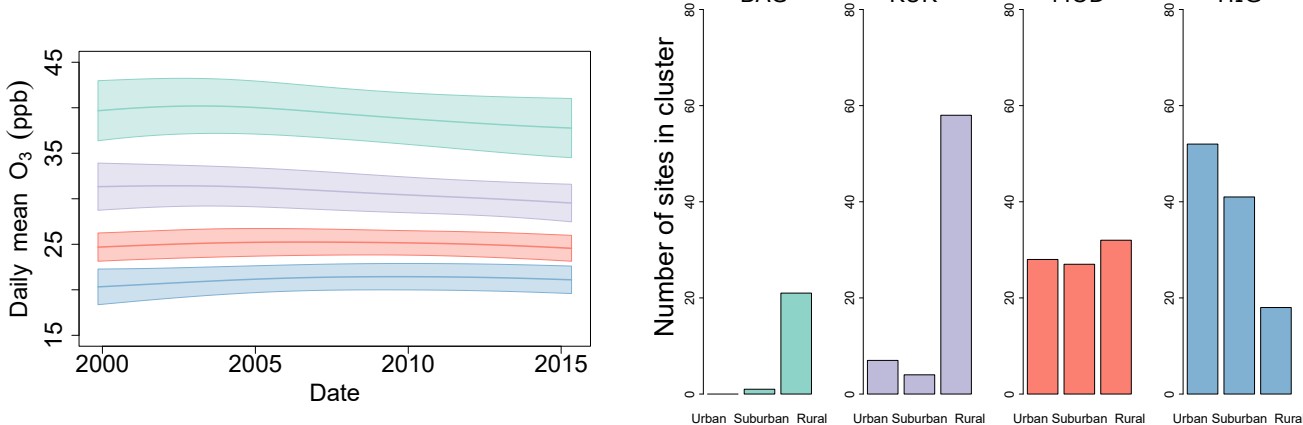

**Figure 3.** Clusters based on daily mean $O_3$ LT(t). Average LT(t) in each cluster with $\pm$ the standard deviation (left) and histograms for the site type included in each cluster.

is high in these sites (Derwent et al., 2004, 2013). In all clusters (except in the "North") the hot summers of 2003 and 2006 are visible in the S(t) signal, which shows that the S(t) signal can capture important events in $O_3$ variability that are driven by seasonal meteorological phenomena.

A two-dimensional classification scheme is achieved by employing the LT(t)- and S(t)-clusters. Our results are in good

agreement with previous classification studies, where by using different $O_3$ metrics similar classifications have been obtained. For instance, the spatial analysis based on gridded $O_3$ data (MDA8) across Europe by Carro-Calvo et al. (2017) resulted in a regional site classification. The gridded data used by Carro-Calvo et al. (2017) were obtained by spatial interpolation leading to a larger and regular geographical coverage compared to the available observations. Compared to Carro-Calvo et al. (2017), similar geographical clusters were identified here, except for the Iberian Peninsula, eastern Europe, northern Scandinavia and

the Balkan states that do not appear as separate clusters in our analysis. This is most probably due to the small number of observational sites in the above regions. More specifically, the sites in the "West" cluster correspond to the Western European and the Iberian Peninsula clusters as extracted by Carro-Calvo et al. (2017), the "North" cluster covers the British Isles, northern Scandinavia, the Baltic region and parts of the north-central Europe clusters of Carro-Calvo et al. (2017). Moreover, the "CentralNorth" cluster includes the north-central, and eastern Europe and parts of the south-central clusters, while the

"CentralSouth" cluster corresponds to the south-central cluster of Carro-Calvo et al. (2017). Finally, the "PoValley" cluster is embedded in the south central cluster of Carro-Calvo et al. (2017). In contrast to our study, $O_3$ concentration during summer have exclusively been used for the cluster analysis by Carro-Calvo et al. (2017), therefore conditions when the correlation of $O_3$ and meteorological variables such as temperature is typically strongest. Four site type clusters were found based on the LT(t) in this study, which are similar to the five site type clusters identified by Lyapina et al. (2016). Similar site classifications

are obtained because the L(t) signal of this study and the mean seasonal and diurnal profiles of Lyapina et al. (2016) both capture the $O_3$ concentration levels distinguishing specific pollution regimes.





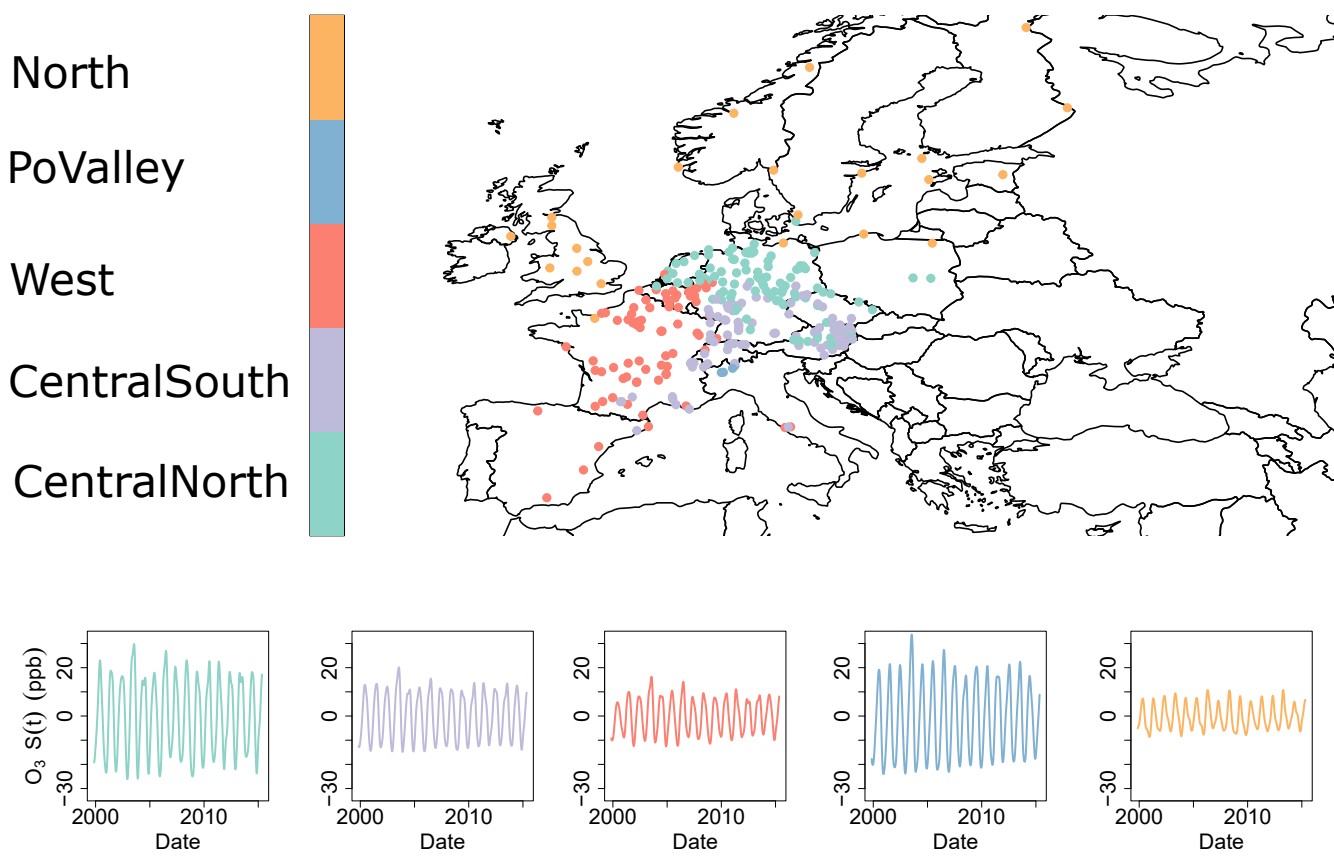

**Figure 4.** Map with the site clusters derived from daily O$_3$ S(t), and average S(t) in each site cluster.

## 4.2 Trends of daily mean O$_3$ concentrations

The daily mean LT(t)- and S(t)-clusters identified in section 4.1 are used for assessment of the temporal trends for the different site types and geographical locations. Overall, decreasing O$_3$ trends are found for rural sites, while there is a tendency for increasing O$_3$ in more polluted urban environments (Fig. 6). The number of sites that belong in each of the identified groups is shown in Table 1. In 64% of all sites significant trends (p-value<0.05) were identified for the daily mean O$_3$; 61% among the significant trends were negative and 39% positive.

Most rural sites – "BAC" and "RUR" – experienced decreasing daily mean O$_3$ concentrations in all regions, as expected following the NO$_x$ and VOC reductions in Europe (Fig. 6). At the "MOD" and "HIG" sites a levelling-off or increase is observed respectively, especially in "CentralNorth", "West" and "North" regions. At the "HIG" sites the positive trends can be partly explained by the increase of NO$_2$ to NO$_x$ ratio, originating from the diesel vehicles, that have increased in the European car fleet (EEA, 2009). In addition, the strong reduction of NO$_x$ concentrations that led to less titration of O$_3$ by NO, could also





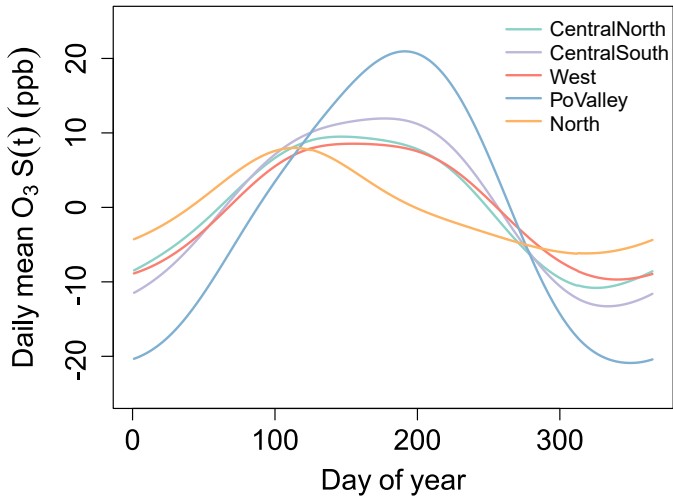

**Figure 5.** Annual cycle of daily mean $O_3$ $S(t)$ for the daily mean $S(t)$ clusters.

explain the positive trends at urban and suburban sites. The late inflection point at urban sites ($LT(t)$ of "HIG" cluster in Fig. 3) can be an additional effect of the reduced titration of $O_3$, which leads to positive trends at the "HIG" sites. Flat trends at central European sites, might partially be explained by the increasing influence of North American and Asian emissions, that have counterbalanced the decrease of European $NO_x$ and VOC concentrations (Derwent et al., 2018; Yan et al., 2018).

5     In agreement with our results, significant decreases of daytime average and summertime mean of MDA8 $O_3$ at European rural sites and small and non-significant downward trends of MDA8 at urban sites have been found previously for the time period 2000-2014 (Chang et al., 2017). Similarly, in a report by EEA (2016) it was found that between 2000 and 2014 annual mean $O_3$ and annual mean MDA8 $O_3$ have been decreasing in rural background sites, while at more polluted sites influenced by nearby man-made precursor emissions, upward trends have been detected.

**Table 1.** Number of sites in each site group based on the $LT(t)$ and $S(t)$ clusters.

| Cluster | BAC | RUR | MOD | HIG | Sum |
|---|---|---|---|---|---|
| CentralNorth | 5 | 8 | 33 | 48 | 94 |
| CentralSouth | 8 | 29 | 27 | 28 | 92 |
| West | 9 | 16 | 21 | 31 | 77 |
| PoValley | 0 | 2 | 2 | 0 | 4 |
| North | 0 | 16 | 4 | 4 | 24 |
| **Sum** | 22 | 71 | 87 | 111 | Total: 291 |





**Figure 6.** Box-plots of de-seasonalized daily mean $O_3$ trends for the LT(t)- and S(t)-clusters. LT(t)-clusters represent a site type classification while the S(t)-clusters a geographical one that is influenced by climatic conditions.

$O_3$ trends at sites in the cluster "North" indicate changes in background $O_3$, especially in the "RUR" ones that are mostly free from local emissions. Here, decreasing trends of daily mean $O_3$ were found in "RUR" sites, while in the "MOD" and "HIG" the trends are slightly increasing. It is interesting to compare the trends in the "North" cluster with the temporal evolution of $O_3$ in Mace Head (remote station in northwestern Ireland), which is representative for inflow of background $O_3$ into Europe. For this reason, we estimated the LT(t) variation of MDA8 $O_3$ and the Theil-Sen trend for the site in Mace Head (Fig. 7). An inflection point was identified in the LT(t) in 2006, i.e. MDA8 $O_3$ has been increasing between 1988 and 2006 and started to slightly decline after 2006. De-seasonalized Theil-Sen trends were estimated 0.08 ppb/year [0.06,0.1] for the first period and -0.04 ppb/year [-0.09,0.02] for the second period.





Similarly, Derwent et al. (2018) have found an increase of $0.34 \pm 0.07$ ppb/year with a deceleration rate after 2007 of $-0.0225 \pm 0.008$ ppb/year$^2$ at the same station, based on a combination of filtered measurement data and modeling output (Lagrangian dispersion). The inflection point in mid-2000s might be the reason for the flat trend of the annual average $O_3$ during 2000s as estimated by Derwent et al. (2013).

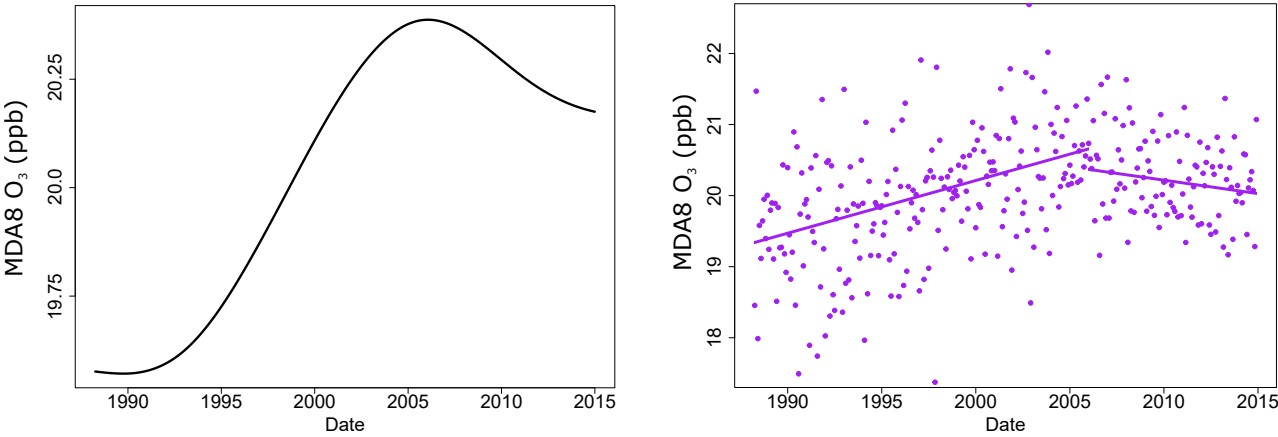

**Figure 7.** MDA8 LT(t) in Mace Head extracted with the EEMD (left) and the corresponding de-seasonalized Theil-Sen trend (right).

### 4.3  Trends of peak $O_3$ concentrations

Peak $O_3$ concentrations in summertime are attributed to increased photochemical production during this time of the year, and the spring maximum in remote locations is linked to increased stratospheric influx as well as hemisphere-wide photochemical production during that season (Holton et al., 1995; Monks, 2000). In this study, significant negative meteo-adjusted MTDM trends were observed at 62% of the sites, while without meteo-adjustment significant negative trends were identified at only 19% of the sites. The higher number of sites with significant trends after the meteo-adjustment indicates the advantage of using meteo-adjusted observations in the trend estimation. This argument is supported in the study by Fleming et al. (2018), where significant negative trends of the 4th highest MDA8 $O_3$ between 2000 and 2014 have been detected at only 18% of the studied sites across Europe, while at a large proportion of sites either weak negative to weak positive or no trends at all were found. The non-significant trends have been attributed by Fleming et al. (2018) to the influence of meteorology which is not considered in their trend estimation.

Trends of meteo-adjusted MTDM are discussed here for the daily mean $O_3$ LT(t)- and S(t)-clusters. MTDM decreased for all site types and regions during the studied period 2000-2015. However, in the "RUR" cluster MTDM showed the strongest decrease among all LT(t)-clusters (Fig. 8). Interestingly, in the "BAC" cluster (especially the "West" cluster) the decrease of MTDM was not so pronounced, likely due to the increase of hemispheric transport of $O_3$ in Europe (Derwent et al., 2007; Vingarzan, 2004). The same pattern was observed at the high alpine site of Jungfraujoch, which is representative for European



continental background O₃ concentrations (Balzani-Lööv et al., 2008; Boleti et al., 2018b). Also, "HIG" sites in "CentralSouth" showed slightly smaller decrease of MTDM compared to the other regions, possibly due to industrialization in the neighboring eastern Europe (Vestreng et al., 2009).

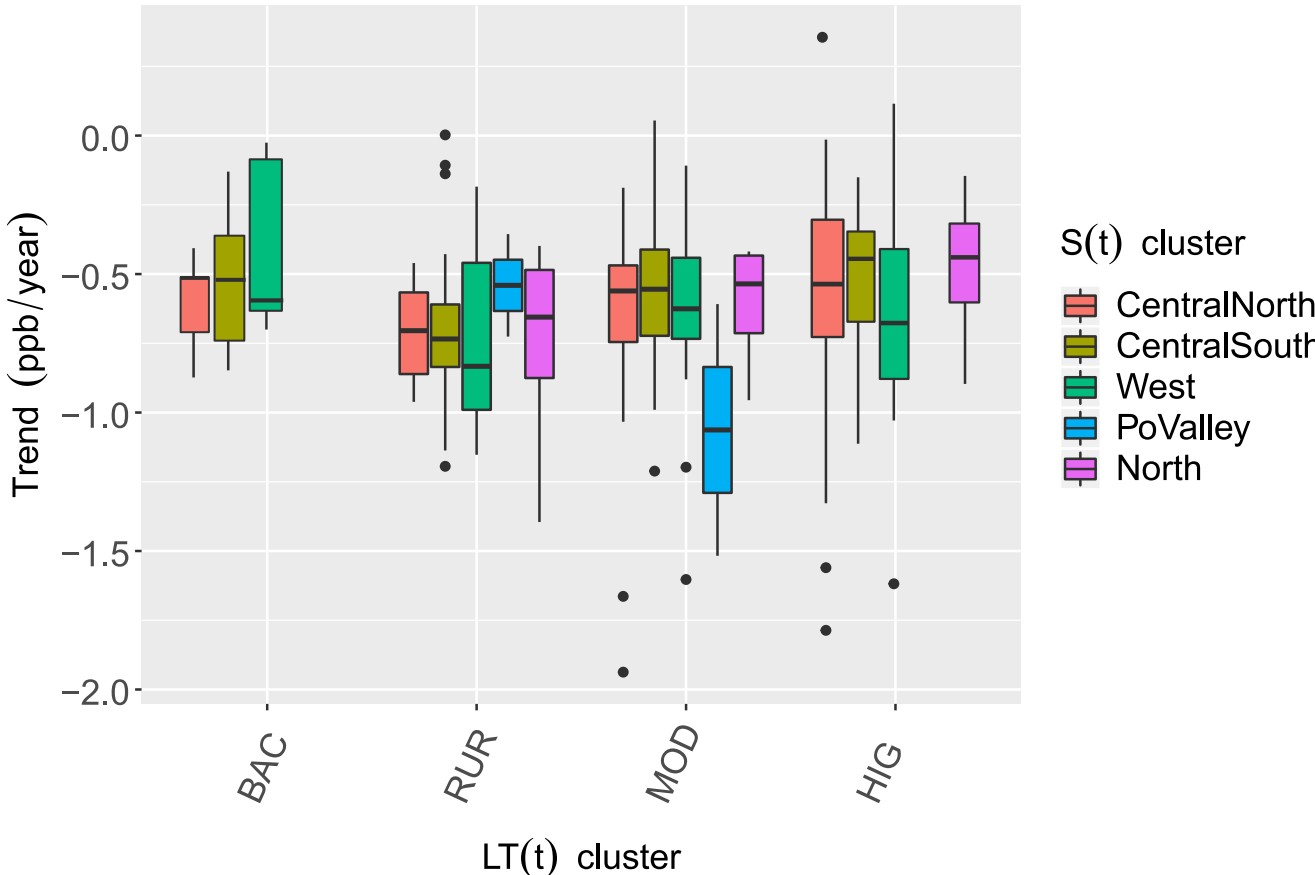

**Figure 8.** Box-plots of meteo-adjusted MTDM trends for the daily mean O₃ LT(t)- and S(t)-clusters. LT(t)-clusters represent a site type classification while the S(t)-clusters a geographical one that is influenced by climatic conditions.

Our results are in line with a modeling sensitivity study, where negative trends of the 95th percentile of O₃ concentrations were found in European rural background sites for the period 1995-2014 (Yan et al., 2018). For the shorter time period between 1995 and 2005, downward trends of measured MTDM have been observed for most parts of Europe as well (in the range [-0.12,-0.55] ppb/year), with the highest decrease in the Czech Republic, UK and the Netherlands (on average -1 ppb/year) and very small (nearly flat) in Switzerland (EEA, 2009). In our study, measured MTDM trends (2000-2015) for these regions are in the same range, i.e. the average decrease was estimated between -0.28 and -0.55 ppb/year. Smaller trends in Switzerland





and central Northeast Germany have been observed by EEA (2009), which agrees with our result for the "CentralSouth" sites that showed the smallest average trend among all clusters. The flat trend in Switzerland during 1995-2005 is probably linked to the disproportional decrease of $NO_x$ and VOCs until beginning of 2000s, when an inflection point has been observed at most polluted sites (Boleti et al., 2018a). In Germany, a mixed behavior was observed by EEA (2009), with the northeastern

part showing a stronger decrease and the central Northeast region a smaller decrease. Similar to our differentiation between the clusters, average MTDM trends within the "CentralNorth" cluster (with northern and northeastern Germany included) were higher and within the "CentralSouth" (covering central parts of Germany) lower.

### 4.4 O$_3$ seasonal cycle trends

Analysis of S(t) allows studying the characteristics of the annual cycle of $O_3$ without influence of short-term meteorological

phenomena and long-term variations. Here, the trends of $S_{max}(t)$, $S_{min}(t)$ and $S_{DoM}(t)$ are presented for the five regions identified based on S(t) as calculated from daily mean $O_3$. A declining amplitude of S(t) and simultaneously a phase shift towards an earlier time in the year can be observed for the 2000 to 2015 period (Table 2).

More specifically, an overall decrease in $O_3$ $S_{max}(t)$ by around 0.05-0.18 ppb/year and a simultaneous increase of $S_{min}(t)$ with a rate of around 0.25 ppb/year was observed for all S(t)-clusters (Fig. 9). However, in the "North" cluster the decrease of

the $S_{max}(t)$ was very small and non-significant, probably due to the pronounced influence of background $O_3$ at these sites. The most pronounced shortening of S(t) amplitude can be seen at the "PoValley" sites, where the downward trend of peak $O_3$ is largest (Fig. 8). The increase in the $S_{min}(t)$ may be partially due to the decreased titration of $O_3$ after reductions of $NO_x$ emissions and probably due to the increased influx of $O_3$ towards north and northwest Europe and more cyclonic activity in the north Atlantic during winter as well (Pausata et al., 2012). A decrease of the 95th percentile and an increase of 5th percentile of

$O_3$ for the period 1995-2014 has been also identified in the EMEP network (rural background sites) (Yan et al., 2018). Lower summertime peaks as a result of decreased photochemical production and higher $O_3$ concentrations during the winter months due to decreased $O_3$ titration have been found in European air masses between 1987 and 2012 (Derwent et al., 2013) as well.

The trend of $O_3$ $S_{DoM}(t)$ is for all regions negative, i.e. the occurrence of the day of maximum $O_3$ has shifted to earlier days within the year with a rate of -0.47 to -1.35 days/year (Table 2, Fig. 10). The observed shift of the day of seasonal maximum

might be linked to the increase of emissions in East Asia that have contributed to increased transport of air pollution to middle- and northern latitudes (Zhang et al., 2016) where the effect on $O_3$ is probably greater due to greater convection, reaction rates and $NO_x$ sensitivity (Derwent et al., 2008; West et al., 2009; Fry et al., 2012; Gupta and Cicerone, 1998). In addition, changes in meteorological factors have affected the seasonal variation of $O_3$. For instance, a similar behavior with an earlier onset of the summer date (the calendar day on which the daily circulation/temperature relationship switches sign) has been observed by

Cassou and Cattiaux (2016) using observational data,while Peña-Ortiz et al. (2015) have found that summer period is extending with a rate of around 2.4 days/decade based on gridded temperature data in Europe. The positive phase of the NAO leads to increased $O_3$ concentrations in Europe through higher westerly winds across the North Atlantic, and enhanced transport of air pollutants from North America to Europe (Creilson et al., 2003). Changes in the NAO have led to increased westerly flow over the North Atlantic during the 1980s and 1990s, which in turn resulted to elevated $O_3$ in northwestern Europe especially



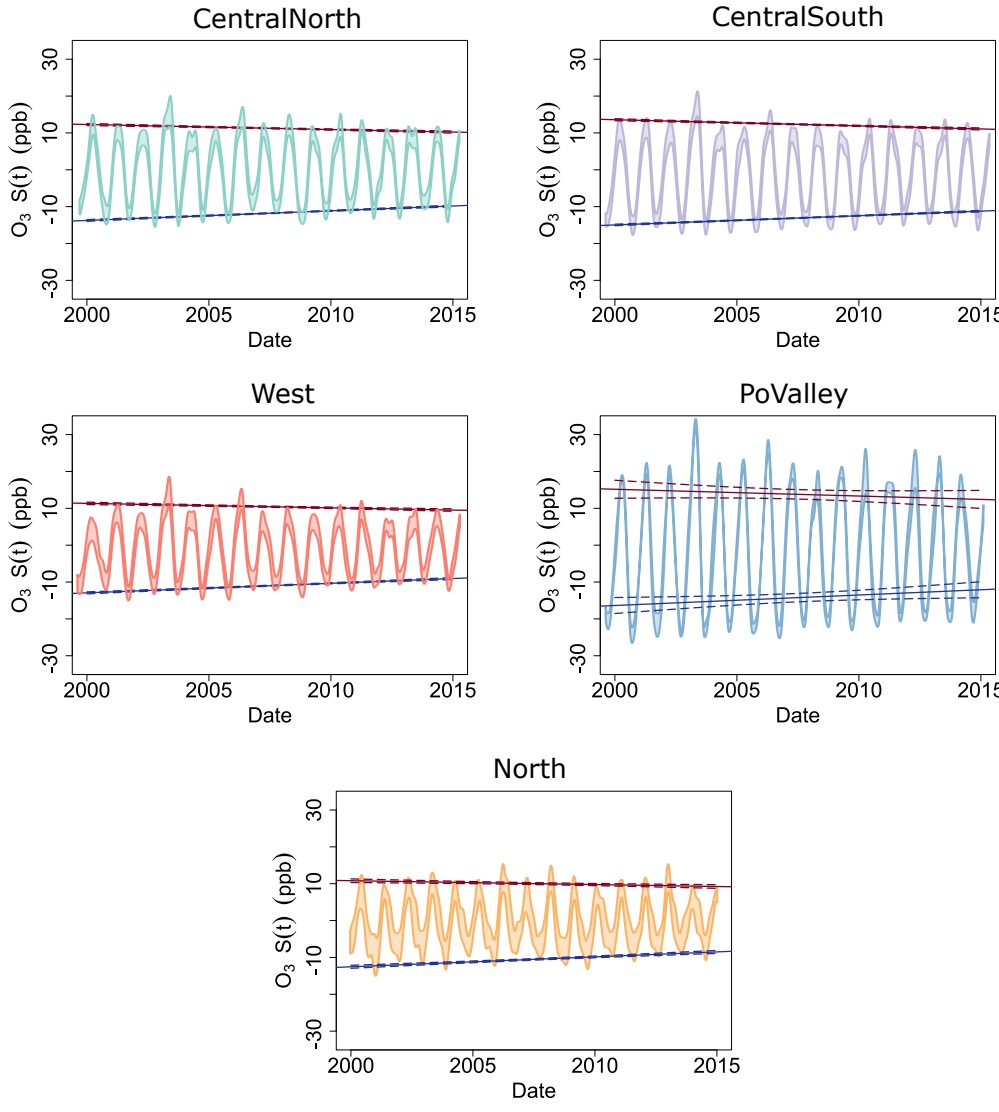

**Figure 9.** Temporal evolution of $S_{max}$ and $S_{min}$ for the daily mean $O_3$ S(t) clusters together with the average S(t) (bands indicate the average $pm$ the standard deviation). Lines show the linear trends of $S_{max}$ and $S_{min}$ and dashed lines the 90% confidence interval.

in winter and spring time (Pausata et al., 2012). The enhanced hemispheric transport of air pollutants from North America to Europe is related to more increased transport through frontal systems (Creilson et al., 2003; Eckhardt et al., 2003). Increased $O_3$ in winter and spring, but not in summer, might lead to a shifting from a pronounced maximum in late summer to a broader spring-summer peak (Cooper et al., 2014). At the "West' sites a slightly stronger shift of the $S_{DoMax}$ was observed compared to other clusters, while at the "North" sites the decrease was the smallest. The early spring maximum in the "North" sites in





April can be explained by higher $NO_x$ that is released from PAN and alkyl nitrates that are produced during winter at northern latitudes (Brice et al., 1984; Bloomer et al., 2010).

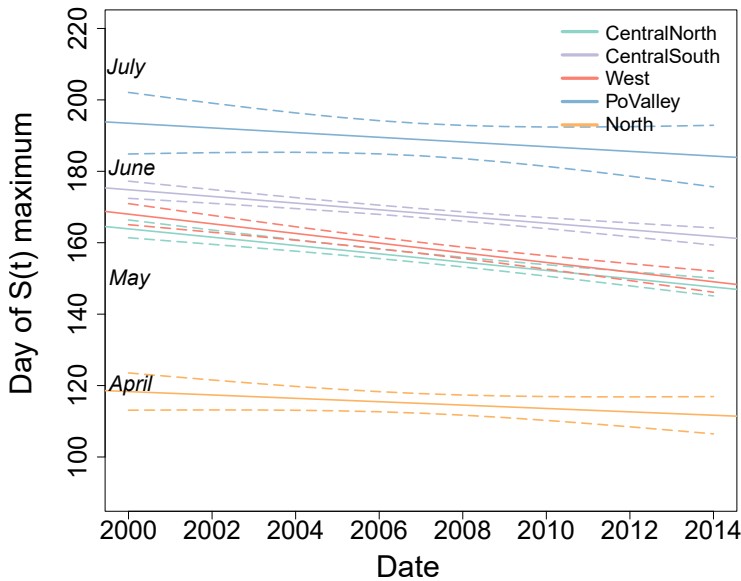

**Figure 10.** Linear trends of the $S_{DoMax}$ for the daily mean S(t) clusters (dashed lines show the 90% confidence interval).

**Table 2.** Linear trends of $S_{max}$, $S_{min}$ and $S_{DoMax}$ for the daily mean $O_3$ S(t) clusters during 2000-2015. $**$ indicate highly significant trend (p-value$< 0.01$), $*$ significant (p-value$< 0.05$) and $-$ indicate non-significant trend (p-value$> 0.05$).

| Daily mean $O_3$ S(t)-Cluster | Trend $S_{max}$ ( ppb/year) | Trend $S_{min}$ ( ppb/year) | Trend $S_{DoMax}$ (days/year) |
|---|---|---|---|
| CentralNorth | -0.14 ± 0.04 (**) | 0.26 ± 0.03 (**) | -1.16 ± 0.18 (**) |
| CentralSouth | -0.17 ± 0.03 (**) | 0.25 ± 0.02 (**) | -0.93 ± 0.17 (**) |
| West | -0.12 ± 0.04 (**) | 0.26 ± 0.03 (**) | -1.35 ± 0.21 (**) |
| PoValley | -0.18 ± 0.6 (-) | 0.30 ± 0.56 (-) | -0.65 ± 0.61 (-) |
| North | -0.05 ± 0.1 (-) | 0.24 ± 0.07 (**) | -0.47 ± 0.38 (-) |

## 4.5 O₃ and temperature relationship

The $O_3$ sensitivity to temperature is a useful metric for validation of precursor reduction scenarios and emission inventories in chemistry-transport models (Oikonomakis et al., 2018). Here, we present the long-term trends of the relationship between the daily maximum $O_3$ concentrations and daily maximum temperature during the warm season from May to September between





2000 and 2015. Daily maximum $O_3$ and temperature are chosen in order to represent peak $O_3$ concentrations formed during the considered days.

Decreasing sensitivity of $O_3$ with respect to temperature was observed during the considered time period in all regions (Fig. 11). Fig. 11 shows the decreasing slopes of linear regression lines of maximum $O_3$ against temperature for successive

5   year groups. The decrease is consistent for all calculated regional clusters except for "North". For most regions in Europe a significant downward trend of the slope of around 0.04-0.05 ppb/K/year was found (Table 3). At "PoValley" sites the decrease was more pronounced (-0.083 ppb/K/year). At the same time the average correlation between $O_3$ and temperature is the highest compared to the other regions, because of large reductions of precursors concentrations in this region which is characterized by high industrial emissions. At the "North" sites the weakest correlation of $O_3$ to temperature was observed and the trend

10   is non-significant. This is expected because at these high latitudes mean temperature is lower compared to other regions in Europe, thus, photochemical production of $O_3$ is weak during the time when $O_3$ typically reaches its maximum concentration.

In relation to the LT(t)-clusters, it was observed that the higher the pollution burden of the site the stronger the trend of $O_3$ to temperature slope (Table 4). As shown here, the "HIG" and "MOD" sites have higher trends compared to the clusters "BAC" and "RUR". Our results are in line with a box-model study that tested the $O_3$–temperature relationship under different $NO_x$

15   level scenarios (Coates et al., 2016). Coates et al. (2016) have shown that at high $NO_x$ conditions $O_3$ increases more strongly with temperature, while the increase is less pronounced when moving to lower $NO_x$ conditions.

**Table 3.** Linear trends of the $O_3$-temperature slope (based on daily maximum values) for the daily mean $O_3$ S(t)-clusters for the time period 2000-2015. ∗∗ indicate highly significant trend (p-value< 0.01), ∗ significant (p-value< 0.05) and − indicate non-significant trend (p-value> 0.05).

| Daily mean $O_3$ S(t)-Cluster | Trend ( ppb/K/year) | Standard deviation | p-value |
|---|---|---|---|
| CentralNorth | -0.042 | 0.003 | ** |
| CentralSouth | -0.04 | 0.003 | ** |
| West | -0.05 | 0.004 | ** |
| PoValley | -0.083 | 0.016 | ** |
| North | -0.016 | 0.013 | - |

**Table 4.** Linear trends of the $O_3$-temperature slope (based on daily maximum values) for the daily mean $O_3$ LT(t)-clusters for the time period 2000-2015. ∗∗ indicate highly significant trend (p-value< 0.01), ∗ significant (p-value< 0.05) and − indicate non-significant trend (p-value> 0.05).

| Daily mean $O_3$ LT(t)-Cluster | Trend ( ppb/K/year) | Standard deviation | p-value |
|---|---|---|---|
| BAC | -0.038 | 0.006 | ** |
| RUR | -0.034 | 0.006 | ** |
| MOD | -0.043 | 0.003 | ** |
| HIG | -0.046 | 0.003 | ** |





**Figure 11.** Linear trend of the slope between $O_3$-temperature daily maximum values for the warm season between May and September. The trends are calculated on the average values for each daily mean S(t) cluster and for the year groups 2000-2005, 2005-2010 and 2010-2015. Points show the mean value for the indicated temperature bin together with the corresponding standard deviation.





## 5 Conclusions

In this study, a classification of 291 sites across Europe was performed for the time period 2000-2015. The clustering algorithm applied on the long-term changes LT(t) and the seasonal cycle S(t) of daily mean O$_3$ resulted in a site type and geographical site classification respectively. Such a classification scheme can be of significant use for O$_3$ trends studies in large spatial domains and in model evaluation studies (e.g. Otero et al., 2018). Our approach captures several features of O$_3$ variations, i.e. pollution level from the L(t)-clustering and influence of the climatic conditions from the S(t)-clustering, and presents a unifying perspective on past studies that report different site type labels based on cluster analysis using different metrics of O$_3$ concentrations. The regional differentiation is hampered by sparse or missing measurement sites in some regions, e.g. eastern Europe or the Balkan peninsula. However, in the last years the number and spatial distribution of sites with longer and more dense measurements has improved.

A trend analysis of de-seasonalized mean O$_3$ concentrations and meteo-adjusted peak O$_3$ concentrations was implemented for the considered sites. By using LT(t)- and S(t)-clusters, patterns of O$_3$ long-term trends across Europe were investigated, based on the multi-dimensional site classification scheme. Long-term trends of de-seasonalized daily mean O$_3$ are decreasing at the rural sites, while in suburban and urban sites they are either stable or slightly increasing. Positive or flat trends indicate that reduction of precursors has been less effective in reducing O$_3$ concentrations in heavily polluted environments. On the other hand, downward trends in peak O$_3$ concentrations were observed in all regions, as a result of precursors emissions reductions. However, peak O$_3$ has been decreasing with the smallest rate at higher altitude sites especially in the western part of Europe due to the influence of background O$_3$ imported from North America and East Asia.

The analysis of S(t) extrema revealed a decrease in summertime maxima and an increase in wintertime minima, pointing to a decreasing amplitude of the seasonal cycle of O$_3$. At the same time the occurrence of the day of maximum has shifted from summer to spring months with a rate of around -0.5 to -1.3 days/year. Changes in the S(t) might be attributed on one hand to the precursors reductions in Europe, and, on the other hand, to changing weather patterns in the northern Atlantic and increase of emissions in southern East Asia.

Finally, the sensitivity of O$_3$ to temperature has weakened since 2000 with a rate of around 0.084 ppb/K/year, i.e. formation of O$_3$ became weaker at high temperature conditions due to decreasing NO$_x$ concentrations. It was shown that differences in changes to this sensitivity across sites are mainly driven by regional meteorological conditions.

*Acknowledgements.* We kindly thank the Swiss Federal Office of Environment for funding this project and Dr. Stephan Henne for extracting the ERA-Interim meteorological variables.





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
