# Peer review of "Temporal and spatial analysis of ozone concentrations in Europe based on time scale decomposition and a multi-clustering approach"

_Atmospheric Chemistry and Physics, 2019_

## Referee Comment (RC1) · Anonymous Referee #1 · 15 Nov 2019

Review of Boleti et al. (ACPD, 2019)

Boleti et al. have used a time series decomposition methodology introduced in their previous papers to extract the long-term, seasonal and short-term components of ozone time series at around 300 measurement sites in Europe. Then they have applied a clustering algorithm separately on the long-term and seasonal components to get a two-dimensional classification of sites, according to the site type (proximity to emission sources) and to the regional characteristics (meteorological influence). Through the combination of such techniques they have gone a step forward compared to previous analyses that resulted in regional site (Carro-Calvo et al., 2017) or site type (Lyapina et

al., 2016) classifications. In addition, grouping the sites according to the two categories as done here simplifies the interpretation of long-term ozone trends.

The manuscript also includes some other powerful analyses, such as the application of a meteorological adjustment technique which has allowed to obtain significantly negative trends of summer peak ozone concentrations at many more sites than in previous trend assessments (e.g. Fleming et al, 2018). Furthermore, through the examination of the seasonal component they document the distinct behavior of some clusters (e.g. maximum ozone occurring earlier in the year over northern Europe than over the Po Valley) as well as a reduction of the amplitude of such cycles and a shift of the day with ozone maximum.

The manuscript represents a substantial contribution to the field and considers related work by including appropriate references. I indeed find it very appropriate for publication in Atmos. Chem. Phys, but at the same time think it should substantially be improved. I have three major concerns. Two of them are related to (i) the choice of daily O3 (instead of MDA8 O3) for the main analyses presented in the manuscript, which has not been justified by the authors, and to (ii) the disconnection between the main text and large parts of the supplementary material (see main comments). My third concern is that the authors should spend time on improving some parts of the manuscript, as seen from the large number of comments included below. I think the manuscript contains a considerable number of inaccuracies, but will fully support the publication once the authors have addressed these comments.

MAIN COMMENTS

(1) The authors address the spatiotemporal variability of daily mean O3 in the main text and leave MDA8 O3 for the supplement. In particular, it is a bit surprising that the daily mean concentrations during the extended summer season are used in section 4.3 (Trends of peak O3 concentrations). Wouldn't it have been more appropriate to use MDA8 O3 at least for that section to focus on the times of the day with the highest

ozone concentrations?

I am not against this choice, but think that the authors should at least justify it. Are they using daily O3 because that simplifies the comparison of most of their results with those of other studies? If that was the case, I would understand that they have preferred sticking to daily O3 in all sections of the main text, just for consistency. Or is there any other reason?

(2) Overall, the main text and the supplement look like two completely separated pieces of work which are not properly linked. The Supplementary Material includes 5 sections and 16 additional Figures, but most of them are neither explained nor referred to from the main text. This is very unpleasant for the reader, who has to look for the appropriate sections and figures in the supplement. Bottom line: the authors should explicitly mention which section/figure of the supplement they are referring to; at the same time, they should not include analyses in the supplement if they do not refer to them from the main text. Here are just some examples:

* Lines 8-11 of page 6: "In addition, a Silhouette width (Sw) analysis is performed to assess the goodness of the clustering (Rousseeuw, 1987). More details about the number of clusters, the goodness of the clustering and the Sw are provided in the supplementary material". Need to refer to some specific sections? Maybe S1-S2?

* Lines 5-6 of page 8: "Here, we present the results of the daily mean LT(t)- and S(t)-clustering; results for the W(t)-clustering and the cluster analysis based on the MDA8 are provided in the supplementary material". Which sections and/or figures of the supplement you are referring to?

* Line 19 of page 9: "The LT(t) signal as derived from the daily mean and MDA8 O3 observations increases" could be changed to The "LT(t) signal as derived from the daily mean (Fig. 3) and MDA8 O3 (Fig. S9) observations increases".

* The results from Sections S3 and S4 (clusters and trends for MDA8 O3) are not

very useful for the reader because there are hardly any specific comments about them in the main text. For instance, are the trends of daily O3 (main text) and MDA8 O3 (supplement) similar? Are the clusters of their L(t), S(t) and W(t) components overall consistent? The authors have two options: either linking the supplement and the main text much better than done now or removing many things from the supplement (e.g. focus only on daily O3 or on MDA8 O3, see previous main comment). I simply think that so much information without some proper explanations in the main text distracts the reader.

SPECIFIC COMMENTS

(1) There are some parts of Section 3 (Methodology) which need further explanations:

1.1. Additional details on the time scale decomposition should be given. For instance, the text around lines 13-15 of page 5 is not very complete: "By adding together the IMFs with frequencies between around 40 days and 3 years we obtain the seasonal variation of O3 ($S(t) = c7 + ... + c10$) and by adding the frequencies that are smaller than 40 days the short-term variation is acquired ($W(t) = c1 + ... + c6$)".

First of all, according to Eq (2), the IMFs (Ci) are time dependent. So I believe it should be "$S(t) = c7(t) + ... + c10(t)$" and "$W(t) = c1(t) + ... + c6(t)$".

The authors should explain where this decomposition (e.g. c7 to c10) and the corresponding time scales (e.g. 40 days to 3 years) come from. If this comes from the previous analyses by Boleti et al (2018) they should explicitly state that.

1.2. The description of the partitioning around medoids (PAM) clustering algorithm used in this study is hard to understand.

For instance, around line 21 of page 5: "PAM is more robust than k-means, because it minimizes the sum of dissimilarities instead of the sum of squared euclidean distances ... Initially, k clusters are generated randomly and the empirical means mk of the euclidean distance between their data points are calculated ...". First the authors say that

PAM does not minimize the sum of squared Euclidean distances but then they mention "euclidean distance" when they refer to mk. I do not get it. By the way, I think it should be "Euclidean" instead of "euclidean".

Around lines 3-4 of page 6: "To identify the optimal number of clusters the k-means algorithm is iteratively executed for a range of k values ...". Now, you are referring to k-means instead of to PAM. Can you please explain all this better? From the present text it is not easy to understand what is different in k-means and PAM.

1.3. Meteorological adjustment (Section 3.4). The authors use GAMS models to fit ozone on a number of variables (eq. 6). Then they follow Barmpadimos et al (2011) to calculate meteo-adjusted ozone as a function of the temporal trend and the residual from the models (eq. 7). Can you please briefly mention how the variable selection is done? Using step-wise regression like in Barmpadimos' work? And what is the overall performance of the meteo adjustement? Similar to that found by previous papers by the same authors for Swiss sites?

(2) The authors should provide further details about the choice, importance and characteristics of the Po Valley cluster (derived from the seasonal component of daily O3, see e.g. Figure 4). Some questions:

2.1. According to that figure and Table 1, the cluster only includes 4 sites. This is too little compared to the other clusters and therefore needs some justification. Would have this cluster appeared if the authors had kept only k=4 instead of k=5 clusters? Even if that was not the case, I understand that it might be appropriate to retain this cluster if the characteristics of this region are very different from those in the surroundings (e.g. elevated emissions and confinement of pollution within a basin with little ventilation, distinct annual cycle as seen from Figure 5).

2.2. around line 29 of page 9: "The sites in "PoValley" display the most pronounced S(t), mainly due to the Mediterranean weather conditions, e.g. high temperatures. At the same time high NOx and VOC emissions in this region leads to higher O3 concentrations". I am not convinced at all with this statement. Note that the amplitude of the S(t) component is remarkably wider both for the Po Valley and the Central North cluster compared to the others (Figure 4). I am surprised at the results for the Central North cluster, where I would expect average ozone concentrations during the warm season (but not in the colder months) to be clearly below those in the Po valley. The authors should explicitly mention this similarity between two apparently very different regions and, if possible, explain why this happens. In other words, are there any reasons why the impact of meteorology and emissions on ozone presents stronger seasonality in these two regions than in others?

In addition, I would remove "Mediterranean weather conditions, e.g. high temperatures", which I find quite vague. I think the expression "Mediterranean weather conditions" is much more appropriate for the coastal sites in Spain, southeastern France and in the proximity of Rome (see Figure 1). I am not sure that "e.g. high temperatures" is appropriate either here because this analysis includes ozone data in all seasons.

(3) In the long paragraph between lines 4-21 of page 10 the authors compare the results to those of previous classifications, namely Carro-Calvo et al. (2017) and Lyapina et al. (2016). See comments:

3.1. The comparison of the results of the S(t) clustering to those by Carro-Calvo is probably too exhaustive. I would simplify it, but this is up to the authors to decide whether they want to do that. Rather than mentioning every single regional difference arising from the comparison of both classifications, I would instead list all the possible reasons why the results of both classifications ae expected to differ. Only some of those reasons are mentioned in the text. Basically, Carro-Calvo used a MDA8 O3 gridded dataset considering only the summer months, while daily O3 at specific sites during the whole year is used here. In addition, Carro-Calvo applied k-means on normalized anomalies while the spatial classification presented here is based on the seasonal component. Finally, the authors are right to indicate that some of the clusters of Carro-Calvo et al. (2017) do not appear here because the former study used gridded data

over locations with few observations, but this explanation is not complete. Note that the final number of clusters will depend on the a priori choices made (e.g. decisions on the number of clusters based on the explained variance achieved, intra-cluster variance or RMSE, minimizing correlations among different clusters, silhouette width, and so on).

3.2. I feel the comparison of the results from the $L(t)$ clustering to those of Lyapina et al. (2016) would benefit from some additional explanations. That work performed two cluster analyses (CA). The first CA used absolute mixing ratio values and resulted in 5 clusters (Table 2 of that paper), while the second CA used normalized mixing rations and yielded 4 clusters (Table 3 of that paper). As it is not straightforward to summarize the description of the clusters in those tables, one could just select one of them (e.g. the first one) and provide some simple explanations. For instance, one could indicate that the results from this study are similar to those of a classification by Lyapina et al. (2016) who found 5 clusters of different type, ranging from urban traffic (equivalent to the "highly polluted" reported here) to rural background.

3.3. It is very good that the authors have acknowledged previous work and compared their results to those studies. Apart from that, either here or somewhere else in the paper, I would emphasize the strength of this work: they authors have clearly gone a step forward compared to those studies because they have provided a two-dimensional classification.

(4) Figure 5 (Annual cycle of daily mean O3 $S(t)$ for the daily mean $S(t)$ clusters) appears on page 12, but I think it is not referenced to from the main text. The figure should be moved to another part of the text (Section 4.4. Ozone seasonal cycles), which would affect the numbering of other figures. Then, in section 4.4, it would be good to mention some of the main features seen from the $S(t)$ component of daily O3 in that figure. For instance, the figure nicely shows that the ozone maxima occurs in summer for the Po Valley cluster and much earlier in the year in the North cluster. This is consistent with previous studies that have reported that both the highest average ozone concentrations and extreme ozone episodes tend to occur over central/southern

Europe during summer and over northern Europe in spring (see e.g. Figs 1 of both Schnell et al., 2015 and Ordóñez et al., 2017). Finally, I would explicitly mention the days of the ozone maxima in each cluster when commenting the trend of DoMax in Table 2.

(5) Section 4.2 is on trends of daily mean ozone, but Figure 7 at the end of that section shows results of MDA8 LT for Mace Head. Why do you use MDA8 instead of daily O3 for that particular figure? Is it just to compare the results with those of Derwent's papers (see first paragraph of page 14)?

(6) As seen from the first paragraph of section 4.3 (trends of peak O3) the main result from that section is that, unlike previous studies like that of Flemming et al. (2018), the meteorological adjustment results in significantly negative trends at many sites. That is a very nice result, but I am not fully convinced with all the interpretations of the trends in the following paragraph. For instance, around lines 17-20 of page 14: "in the "BAC" cluster (especially the "West" cluster) the decrease of MTDM was not so pronounced, likely due to the increase of hemispheric transport of O3 in Europe (Derwent et al., 2007; Vingarzan, 2004)". However, those papers roughly cover the first halve of the period of analysis, where I agree that might have been the case (see e.g. Figure 7 for a different metric at Mace Head). Moreover, a few lines below (lines 1-3 of page 15) they claim that there might be some connection between the industrialization of Eastern Europe and the trends in some clusters (lines 1-3 of page 15).

I admit that these interpretations are plausible and that the authors have been reasonably careful with their statements, but I would add a short sentence to mention that some more dedicated analyses (e.g. modelling studies) would be needed to investigate the reasons for such trends. I fully understand that such analyses are out of the scope of this paper.

(7) I also like the idea of examining the seasonal cycles of O3 in Section 4.4 and the results presented there are relevant. However, I am not convinced about some of the

explanations given there as there are some inaccuracies. In addition, I am not happy at all with the writing and believe that this section has been written in a rush. There are so many inaccuracies and corrections to make (some of them included in the technical corrections section) that it very hard to focus on the science. Examples:

* Lines 17-19 of page 16: "The increase in the S min (t) may be partially due to the . . . and probably due to the increased influx of O3 towards north and northwest Europe and more cyclonic activity in the North Atlantic during winter as well (Pausata et al., 2012)". Apart from improving the writing (too many "ands" within the same sentence), I am not convinced at the explanations regarding Pausata's paper. What you do mean by increased influx and cyclonic activity? Are such things really mentioned that way in that paper? If so please explain this better. As far as I remember, that work simply suggested that the increasing baseline ozone in western and northern Europe during the 1990s could be associated with a prevailing NAO+ phase during that period.

* Lines 24-27 of page 16: "The observed shift of the day of seasonal maximum might be linked to the increase of emissions in East Asia that have contributed to increased transport of air pollution to middle-and northern latitudes (Zhang et al., 2016) where the effect on O3 is probably greater due to greater convection, reaction rates and NOx sensitivity [some refs.]. . .". Need to completely rewrite this sentence because it is hard to understand. I assume that the strong convection takes place in East/Southeast Asia instead of at mid/north latitudes as it reads now from this sentence. In addition, the word "greater" is repeated within the same line.

* Line 31 – 33: "The positive phase of the NAO leads to increased O3 concentrations in Europe through higher westerly winds across the North Atlantic, and enhanced transport of air pollutants from North America to Europe (Creilson et al., 2003)." All this looks a bit redundant. Do you simply mean that "The positive phase of the NAO leads to increased O3 concentrations in Europe through enhanced transport of ozone and precursors across the North Atlantic from North America to Europe (Creilson et al., 2003)"?

TECNICAL COMMENTS AND CORRECTIONS

* Line 3 of page 1 (abstract): "the effect of these reductions on ozone is investigated by analyzing surface measurements of ozone". Change the second "ozone" to "this pollutant" just to avoid redundancies.

* Lines 13-14 of page 1: "The effect of hemispheric transport of ozone can be seen either in regions affected by synoptic patterns in the northern Atlantic or at sites located at remote high altitude locations". I do not consider this as an appropriate sentence for the abstract. The manuscript includes some references on the impact of long-range transport and changing weather patterns (e.g. impact of the NAO), but it does not provide any supportive evidence of the relevance of such processes.

* Lines 17-18 of page 1: "while seasonal cycle trends and changes in the sensitivity of ozone to temperature are driven by regional climatic conditions". I would tone down this statement. Honestly, I do not think that the manuscript proves that this impact is larger than that of the varying rates of ozone precursor emission reductions over the different regions.

* Last line of page 2 (Introduction): "For instance, in the U.S., O3 climate penalty – defined as the slope of the O3 versus temperature relationship – dropped from 3.2 ppbv/C before 2002 to 2.2 ppbv/C after 2002 as a result of NOx emission reductions (Bloomer et al., 2009)". One could add a reference to Colette et al. (2015), who analysed chemistry-transport and climate-chemistry model projections to asses the impact of climate change on this climate penalty over Europe by the turn of the century.

* First line of page 4: "during May and September" –> between May and September

* 5th line of page 4: Did you use surface pressure or sea level pressure (SLP)?

* Line 21 of page 5: Move "e.g." to the beginning in "(Lyapina et al., 2016, e.g. )".

* Line 17 of page 6: where yd (t) the de-seasonalized –> where yd (t) is the de-seasonalized

**[ACPD](ACPD)**

Interactive
comment

* Line 19 of page 6: "of the" can be removed from "de-seasonalized concentrations of the yd (t)".

* Line 25 of page 6: "because" is repeated within the same sentence.

* Lines 4-5 of page 7: "For the GAMs, the following meteorological variables were used". I would remove "meteorological". Reason: the Julian day, which is not a meteorological variable, is also included in the model.

* Line 6 of page 7: Again "daily mean surface pressure". Do you mean SLP?

* Line 7 of page 7: No need to spell out CAPE again.

* Line 8 of page 9: "mostly located at higher altitudes". Higher than what? "Higher" could be changed to something like "relatively high" or "elevated".

* Lines 9-10 of page 11: "the positive trends can be partly explained by . . . originating from the diesel vehicles". Change to "the positive trends could partly be explained by . . . originating from the proliferation of diesel vehicles"

* Line 6 of page 12: "rural sites and small and non-significant" –> "rural sites as well as small and non-significant"

* Figure 6: Please indicate what the box-plots indicate (i.e. median, edges of the boxes: 25-27th percentiles, whiskers related to interquartile range or to some specific percentiles, etc.).

* Lines 7-8 of page 13: "Trends were estimated 0.08 ppb/year [0.06,0.1] for the first period and -0.04 ppb/year [-0.09,0.02] for the second period". Lines 1-2 of page 14: "Derwent et al. (2018) have found an increase of 0.34 $\pm$ 0.07 ppb/year with a deceleration rate after 2007 of -0.0225$\pm$ 0.008 ppb/year". If possible, indicate if the uncertainty estimates correspond to the 95% confidence intervals or to something else. There are other parts of the text where this is not clear.

* Section 4.2 is on the trends of daily mean ozone, but Figure 7 at the end of that

section shows results of MDA8 LT for Mace Head. The results look convincing, but why do the authors use MDA8 instead of daily O3 for that particular figure? Is this just to compare the results with those of Derwent's papers (see first paragraph of page 14)?

* Lines 6-7 of page 14: The word "increased" can be removed from both lines.

* Lines 1 and 5 of page 16: "central Northeast Germany" and "central Northeast region". Do you mean "central and northeast"?

* Line 28 of page 16: "meteorological factors have affected" –> "meteorological factors may have affected".

* Line 30 of page 16: Add space before while in "data,while".

* Line 34 of page 16: "resulted to"–> "resulted in" (this should be changed somewhere in the supplement too).

* Caption of Figure 9 on page 17: Change "pm" to "+/-" in "average pm the standard deviation".

* Line 2 of page 17: "more increased" –> "increased"

* Lines between pages 17 and 18: The early spring maximum in the "North" sites in April can be explained by higher NOx that is released from PAN and . . .". What do you mean by "higher NOx? Higher than what? Do you mean something like elevated NOx? By the way, why don't you refer to Figure 5 here (see comment above)?

* Table 2 on page 18: According to the methods section, one should write "SDoM". Need to change that in the caption and header of last column.

* Lines 6-9 of page 19: "At PoValley sites the decrease was more pronounced (-0.083 ppb/K/year). At the same time the average correlation between O3 and temperature is the highest compared to the other regions, because of large reductions of precursors concentrations in this region which is characterized by high industrial emissions". I assume you mean something like "At PoValley sites the decrease was more pronounced (-0.083 ppb/K/year) because of large reductions of precursor concentrations in this region which is characterized by high industrial emissions. Note that the average correlation between O3 and temperature in that cluster is the highest compared to the other regions".

* Lines 9-11 of page 19: Regarding the low correlations between O3 and T in the North cluster I would also mention the low temperature ranges observed there compared to the other clusters (Fig 11).

* Lines 12-16 of page 19. Discussion of the stronger trends of the O3-T relationship for the more polluted LT(t) clusters: "Our results are in line with a box-model study that tested the O3–temperature relationship under different NOx level scenarios (Coates et al., 2016). Coates et al. (2016) have shown that at high NOx conditions O3 increases more strongly with temperature, while the increase is less pronounced when moving to lower NOx conditions". As mentioned for the Po Valley S(t) cluster (see a couple of comments above), I don't see a clear relationship. These references are related to the strength of the O3-T relationship but not to the trend of that relationship. I assume that you mean that regional ozone production has mainly decreased at the most polluted locations, due to considerable reductions of precursor emissions there. Need to rewrite.

* One can remove the columns "standard deviation" in Tables 3 and 4. The p-values should suffice.

* In different parts of the text, the authors indicate that the S(t) clusters represent the "climatic conditions". I would add the word "regional" to clearly indicate that this is indeed a geographical classification that clearly reflects the regional climate conditions. This can be done in different parts of the text. Here I simply include an example for lines 5-6 of page 21: "Our approach captures several features of O3 variations, i.e. pollution level from the L(t)-clustering and influence of the climatic conditions from the S(t)-clustering". I would change the end of this sentence to "influence of the regional

climate conditions from the S(t)-clustering".

* Lines 17-18 of page 21 (Conclusions): "peak O3 has been decreasing with the smallest rate at higher altitude sites especially in the western part of Europe due to the influence of background O3 imported from North America and East Asia". Are the evidences for this long-range-transport influence so clear? If not I would add the word "possibly" before "due to".

* Line 24 of page 21: "the sensitivity of O3 to temperature has weakened since 2000 with a rate of around 0.084 ppb/K/year". It should be considerably less than that because that value is only found for the 4 sites in the Po Valley cluster (see Tables 3 and 4).

* Lines 25-26 of page 21 (about the decreasing O3-T slope): "It was shown that differences in changes to this sensitivity across sites are mainly driven by regional meteorological conditions". I do not see any proof of this in the manuscript. It might well be related to varying rates of reductions of precursor emission across the different regions. I have a similar comment about this in the abstract so both of them can be addressed at the same time.

* The references Boleti et al. (2018a) and Boleti et al. (2018b) should be changed to Boleti et al. (2018) and Boleti et al. (2019), respectively. The second paper has been published and should be updated in the reference list.

* Figures S1 & S2: Can you please enlarge fonts. They are too small and very hard to read.

* Lines 2-3 of page 3 in Supplement: "objects have a low similarity with the rest objects" –> "objects have low similarity with the rest of the objects"

* Figure S3: Even when there are some explanations on the previous page, most readers are not very familiar with the concept of silhouette width. In the figure caption I would indicate (1) that each horizontal bar represents the silhouette width for a particular site in a given cluster and (2) that this parameter is clearly positive for most sites.

* Figure S5: I assume this is for W(t) instead of for S(t).

* Section S3 (Additional information on clusters): What is the use of this section if you hardly provide any comments e.g. about the MDA8 O3 clusters in the main manuscript? As indicated above, the main text and most of the supplement seem detached from each other.

* Lines 2-4 of page 10 in supplement: "In this section we present more detailed information about the clusters extracted from the daily mean and MDA8 O3 LT(t), S(t) and W(t)". Please refer to Figures S9-S11 there.

* Caption of Figure S9: Need to remove "Map indicating the sites that belong in each cluster and average LT(t) in each cluster with the standard deviation of the sites that have SW>0".

* Captions of Figures S10-S12: Need to use subscript for W in "SW" (silhouette width, defined in section S2).

* Lines 4-5 of page 14 in supplement: "The level off or small increase in the HighPoll stations can be attributed to the smaller rate of reduction of VOCs, which resulted to reduced titration of O3 by NO". I agree with the reduced titration due to the decreases of NOx emissions, but do you have any evidence about the smaller rate of reduction of VOCs? If so one should provide a reference. A simpler explanation might be a change of chemical regimes, i.e. in the sensitivity of O3 production to NOx and VOCs.

* Line 6 of page 14 in supplement: Change "(Fig. S14" to "(Fig. S14)". Then remove "respectively" because it is not needed there.

* Lines 10-11 of page 14 in supplement: "Here, the sites with negative SW that were not considered in the discussion of the trends are presented. In the LT(t)-clustering four sites with negative SW were identified (Fig. S15), in the S(t) 26 sites (Fig. S16)

and in the W(t) 24 sites". It is unclear whether you are talking about the clusters of daily O3 or MDA8 O3.

* Caption of Figure S15 is not complete. Need to indicate the meaning of the different lines/shading.

* Caption of Figure S16 is not complete either. For instance you don't show the "clusters average S(t)" as you claim because there is some spread in the figure. Probably mean +/- standard deviation?

REFERENCES

Colette et al.: Is the ozone climate penalty robust in Europe? Environ. Res. Lett. 10 (2015), 084015, doi:10.1088/1748-9326/10/8/084015.

Ordóñez, C., Barriopedro, D., García-Herrera, R., Sousa, P. M., and Schnell, J. L.: Regional responses of surface ozone in Europe to the location of high-latitude blocks and subtropical ridges, Atmos. Chem. Phys., 17, 3111–3131, https://doi.org/10.5194/acp-17-3111-2017, 2017.

Schnell, J. L., Prather, M. J., Josse, B., Naik, V., Horowitz, L. W., Cameron-Smith, P., Bergmann, D., Zeng, G., Plummer, D. A., Sudo, K., Nagashima, T., Shindell, D. T., Faluvegi, G., and Strode, S. A.: Use of North American and European air quality networks to evaluate global chemistry–climate modeling of surface ozone, Atmos. Chem. Phys., 15, 10581–10596, https://doi.org/10.5194/acp-15-10581-2015, 2015.

---

## Referee Comment (RC2) · Anonymous Referee #2 · 3 Dec 2019

The manuscript presented by Boleti et al. examines trends on surface ozone concentrations across a number of stations over Europe for the period 2000-2015. They use a time-scale decomposition to analyse long-term (LT), seasonal (S) and short-term (W) variations. Then, they apply a clustering technique and they finally calculate the trends in the clusters obtained. In addition, they analyse the ozone-temperature relationship over the different clusters and sub-periods. Their classification is consistent with previous studies and their results show a general decreasing in the ozone concentrations, mostly in the ozone peaks. In addition they find a reduced sensitivity in the ozone-temperature relationship over most of the clusters defined.

Overall, I found the manuscript very interesting and complete. The methodology applied is robust and consistent, as well as the results presented. However, I also think that there are some parts in the current version that should be improved in order to be published, in particular the methods sections (see my comments below). In my opinion the manuscript might be a good contribution to *Atmos. Chem. Phys* and the scientific community. Therefore, I would be happy to support the publication of the present manuscript after addressing some comments, which I consider that would be useful to improve it.

I have a few general comments and some specific comments:

**General comments**

There are some parts in the methods section that are not very clear, and in my opinion this section is essential to follow the manuscript. Therefore I have some comments and questions that I would like to ask the authors:

Section 3.1. Time scale decomposition (page5): The authors should explain in more detail the IMF. How the number of coefficients ($c_j$) is selected? The authors say " By adding together IMFs with frequencies around 40 days and 3 years we obtain the seasonal variation of O3 ($c_1+…c_{10}$)", but why 40 days and 3 years? Is this based on the previous study from Boleti et al.2018? I think that this information should be included in order to help the reader to better understand the methodology.

Sections 3.3-3.4 Long-term trends (page 6, 7):

I understand that for the peaks of O3 metrics the method explain in section 3.3 cannot be applied. But, it would possible to use the same method, i.e. GAMs models also for daily mean and MDA8 O3, wouldn't it?

Why the authors define the warm season as May-September? Why April is not included? I think this should be further clarified, since usually ozone season ranges from April to September (e.g. EEA, 2019, Fleming et al. 2018)

Reading the modelling part (section 3.3, 3.4, page 6) is not clear the input data to calculate the trends, e.g. the GAM models are fitted to each cluster that contains a number of stations, so the models are applied individually to each station, am I right?

Regarding the analysis of seasonal cycle of O3, why do the authors chose the mean of O3 and not the MDA8O3?

**Specific comments**

L26-30 of page 3. The authors applied a filter to obtain the time series, and only those with a maximum of 15% of missing values and maximum of 120 consecutive days are used. Is this 15% applied to whole period (16 years) or each year? And the consecutive days? I assume that they refer those 120 consecutive days in one year, is that correct? Can the authors clarify this?

L1 of page 5. I would add that the clusters are identified by using each component L(t), S(t), W(t) **separately** to the algorithm.

L4 of page 8. What are the temperature ranges considered?

L4 of page 9. Why do the authors leave the results of MDA8O3 in the supplement and the results if the O3 in the main text? Wouldn't it be more interesting to see the results for MDA8O3?

L19-23 of page 9. Please refer to figure 3.

L25 of page9. Just a comment regarding Fig.4. the colours for "Po Valley" and "Central South" maybe could be changed, they are quite similar and it is hardly to distinguish the stations that belong to each cluster.

L4-L20 of page 10. Should Figure 5 be referred here? I couldn't find any reference to Fig.5.

L3 of page 11. "decreasing O3 trends", maybe it should be specified "decreasing daily O3 means".

L4 of page 11. In my opinion, the table 1 with the number of stations should be introduced before (e.g. when presenting the clusters).

L9 of page 14. Where these percentages 62% and 18% came from exactly? Is there any figure to support this? This question is in the line of one my previous comment (i.e. how the models are fitted).

L10 of page 19. In the North sites the variability of temperature is lower and O3 is also more influenced by transport.

L24-26 of page 21. The authors attributed the decreasing O3-Temperature to NOx reductions, and this is likely one of reasons (but it is not showing here) and then, they mention that "changes in the sensitivity across sites are mainly driven by regional meteorological conditions", so what about the NOx emission reductions just mentioned? I think this last paragraph is important and it must be rewritten.

**References**

EEA: Air quality in Europe—2019. https://www.eea.europa.eu/publications/air-quality-in-europe-2019

Fleming, Z. L., Doherty, R. M., Schneidemesser, E. V., Malley, C. S., Cooper, O. R., Pinto, J. P., Colette, A., Xu, X., Simpson, D., Schultz, M. G., Lefohn, A. S., Hamad, S., Moolla, R., and Solberg, S.: Tropospheric Ozone Assessment Report : Present-day ozone distribution and trends relevant to human health, Elementa: Science of the Anthropocene, 6, doi:https://doi.org/10.1525/elementa.273, 2018.

---

## Author Comment (AC1) · 19 Feb 2020

**Response to reviewer comments: "Temporal and spatial analysis of ozone concentrations in Europe based on time scale decomposition and a multi-clustering approach"**

Eirini Boleti, Christoph Hueglin, Stuart Grange, Andre Prevot, Satoshi Takahama

**Response to reviewer 1**

Boleti et al. have used a time series decomposition methodology introduced in their previous papers to extract the long-term, seasonal and short-term components of ozone time series at around 300 measurement sites in Europe. Then they have applied a clustering algorithm separately on the long-term and seasonal components to get a two-dimensional classification of sites, according to the site type (proximity to emission sources) and to the regional characteristics (meteorological influence). Through the combination of such techniques they have gone a step forward compared to previous analyses that resulted in regional site (Carro-Calvo et al., 2017) or site type (Lyapina et al., 2016) classifications. In addition, grouping the sites according to the two categories as done here simplifies the interpretation of long-term ozone trends. The manuscript also includes some other powerful analyses, such as the application of a meteorological adjustment technique which has allowed to obtain significantly negative trends of summer peak ozone concentrations at many more sites than in previous

trend assessments (e.g. Fleming et al, 2018). Furthermore, through the examination of the seasonal component they document the distinct behavior of some clusters (e.g. maximum ozone occurring earlier in the year over northern Europe than over the Po Valley) as well as a reduction of the amplitude of such cycles and a shift of the day with ozone maximum. The manuscript represents a substantial contribution to the field and considers related work by including appropriate references. I indeed find it very appropriate for publication in Atmos. Chem. Phys, but at the same time think it should substantially be improved. I have three major concerns. Two of them are related to (i) the choice of daily O3 (instead of MDA8 O3) for the main analyses presented in the manuscript, which has not been justified by the authors, and to (ii) the disconnection between the main text and large parts of the supplementary material (see main comments). My third concern is that the authors should spend time on improving some parts of the manuscript, as seen from the large number of comments included below. I think the manuscript contains a considerable number of inaccuracies, but will fully support the publication once the authors have addressed these comments.

**Main Comments**

1. The authors address the spatiotemporal variability of daily mean O3 in the main text and leave MDA8 O3 for the supplement. In particular, it is a bit surprising that the daily mean concentrations during the extended summer season are used in section 4.3 (Trends of peak O3 concentrations). Wouldn't it have been more appropriate to use MDA8 O3 at least for that section to focus on the times of the day with the highest ozone concentrations? I am not against this choice, but think that the authors should at least justify it. Are they using daily O3 because that simplifies the comparison of most of their results with those of other studies? If that was the case, I would understand that they have preferred sticking to daily O3 in all sections of the main text, just for consistency. Or is there any other reason?

This is true, the reason we use daily mean O3 in all sections is to be consistent throughout the manuscript and be able to compare within the paper. On the other hand, the clusters based on O3 daily mean and MDA8 are rather similar, thus, the choice of clusters between the two metrics does not affect the conclusions. As the referee mentions here, it is appropriate to use the same clusters throughout the paper, in order to compare and discuss the results. It was shown in the section 4.1 of the manuscript that the O3 daily mean depicts the main influencing factors for O3 trends, proximity to emission sources and meteorological conditions, therefore, it is appropriate to study different trend metrics based on the O3 daily mean clusters. We added the following sentences to clarify this issue in the manuscript: "The clusters based on daily mean and MDA8 $O_3$ are similar and the choice of clusters based on these two metrics does not affect the conclusions. The daily mean $O_3$ clusters depict the main influencing factors for $O_3$ trends, i.e. proximity to emission sources of precursors and meteorological conditions."

2. Overall, the main text and the supplement look like two completely separated pieces of work which are not properly linked. The Supplementary Material includes 5 sections and 16 additional Figures, but most of them are neither explained nor referred to from the main text. This is very unpleasant for the reader, who has to look for the appropriate sections and figures in the supplement. Bottom line: the authors should explicitly mention which section/figure of the supplement they are referring to; at the same time, they should not include analyses in the supplement if they do not refer to them from the main text.

Here are just some examples: Lines 8-11 of page 6: "In addition, a Silhouette width (Sw) analysis is performed to assess the goodness of the clustering (Rousseeuw, 1987). More details about the number of clusters, the goodness of the clustering and the Sw are provided in the supplementary material". Need to refer to some specific sections? Maybe S1-S2?

Lines 5-6 of page 8: "Here, we present the results of the daily mean LT(t)- and

S(t)-clustering; results for the W(t)-clustering and the cluster analysis based on the MDA8 are provided in the supplementary material". Which sections and/or figures of the supplement you are referring to?

Line 19 of page 9: "The LT(t) signal as derived from the daily mean and MDA8 O3 observations increases" could be changed to The "LT(t) signal as derived from the daily mean (Fig. 3) and MDA8 O3 (Fig. S9) observations increases".

The results from Sections S3 and S4 (clusters and trends for MDA8 O3) are not very useful for the reader because there are hardly any specific comments about them in the main text. For instance, are the trends of daily O3 (main text) and MDA8 O3 (supplement) similar? Are the clusters of their L(t), S(t) and W(t) components overall consistent? The authors have two options: either linking the supplement and the main text much better than done now or removing many things from the supplement (e.g. focus only on daily O3 or on MDA8 O3, see previous main comment). I simply think that so much information without some proper explanations in the main text distracts the reader.

That is true, some sections in the supplementary are not clearly linked to the main manuscript. We corrected the points indicated by the referee, by referring to the corresponding section of the supplementary material where is needed in the manuscript :
Lines 8-11 of page 6: "In addition, a Silhouette width (Sw) analysis is performed to assess the goodness of the clustering (Rousseeuw, 1987). More details about the number of clusters, the goodness of the clustering and the Sw are provided in the supplementary material (Sections S1 and S2)". Lines 5-6 of page 8: "Here, we present the results of the daily mean LT(t)- and S(t)-clustering; results for the W(t)-25clustering and the cluster analysis based on the MDA8 are provided in the supplementary material (Section S3)". Line 19 of page 9: "The LT(t) signal as derived from the daily mean and MDA8 O3 observations increases" has now changed to The "LT(t) signal as derived from the daily mean (Fig. 3) and MDA8

O3 (Fig. S9) observations increases". Sections S3 and S4 are now mentioned in the text, so that the reader can refer to the corresponding sections of the supplementary material.

**Specific comments**

1. There are some parts of Section 3 (Methodology) which need further explanations:

   (a) Additional details on the time scale decomposition should be given. For instance, the text around lines 13-15 of page 5 is not very complete: "By adding together the IMFs with frequencies between around 40 days and 3 years we obtain the seasonal variation of O3 ($S(t) = c_7 + ... + c_{10}$) and by adding the frequencies that are smaller than 40 days the short-term variation is acquired ($W(t) = c_1 + ... + c_6$)". First of all, according to Eq (2), the IMFs ($C_i$) are time dependent. So I believe it should be "$S(t) = c_7(t) + ... + c_{10}(t)$" and "$W(t) = c_1(t) + ... + c_6(t)$". The authors should explain where this decomposition (e.g. $c_7$ to $c_{10}$) and the corresponding time scales (e.g. 40 days to 3 years) come from. If this comes from the previous analyses by Boleti et al (2018) they should explicitly state that.

   Indeed, the discussion on the choice of the IMFs for the seasonal and short term variations is in the publication by **?**. This is now made more clear in the manuscript with the following sentence, as an extended explanation on this approach is not in the scope of this study. "A more detailed discussion on the choice of the IMFs for the seasonal and short term variations can be found in **?**. " The equation for $S(t)$ has been also corrected.

   (b) The description of the partitioning around medoids (PAM) clustering algorithm used in this study is hard to understand. For instance, around line 21 of page 5: "PAM is more robust than k-means, because it minimizes the

sum of dissimilarities instead of the sum of squared euclidean distances. .
. Initially, k clusters are generated randomly and the empirical means mk of
the euclidean distance between their data points are calculated." First the
authors say that PAM does not minimize the sum of squared Euclidean dis-
tances but then they mention "euclidean distance" when they refer to mk.
I do not get it. By the way, I think it should be "Euclidean" instead of "eu-
clidean". Around lines 3-4 of page 6: "To identify the optimal number of
clusters the k-means algorithm is iteratively executed for a range of k values
. . .". Now, you are referring to k-means instead of to PAM. Can you please
explain all this better? From the present text it is not easy to understand
what is different in k-means and PAM.

The main difference between k-means and PAM is that k-means uses cen-
troids, while PAM uses the medoids, but they both use the Euclidean dis-
tance as a measure of difference between the data points. Indeed, this was
not clear enough in the manuscript. We changed the text to the following
statement: "PAM is more robust than k-means and less sensitive to outliers,
because it uses medoids instead of centroids ."

(c) Meteorological adjustment (Section 3.4). The authors use GAMS models to
fit ozone on a number of variables (eq. 6). Then they follow Barmpadimos
et al (2011) to calculate meteo-adjusted ozone as a function of the temporal
trend and the residual from the models (eq. 7). Can you please briefly
mention how the variable selection is done? Using step-wise regression like
in Barmpadimos' work? And what is the overall performance of the meteo
adjustement? Similar to that found by previous papers by the same authors
for Swiss sites?

Variable selection was not performed in this study, but the meteorological
variables used here are the ones found by Boleti et al (2019) to be the most
important for O3 maximum concentrations. Nevertheless, we have now im-
proved the manuscript to make this point more clear to the reader with the

following statement: "The above explanatory meteorological variables are the ones that were most often selected in the Swiss sites by the meteorological variable selection performed by **?**."

2. The authors should provide further details about the choice, importance and characteristics of the Po Valley cluster (derived from the seasonal component of daily O3, see e.g. Figure 4). Some questions:

   (a) According to that figure and Table 1, the cluster only includes 4 sites. This is too little compared to the other clusters and therefore needs some justification. Would have this cluster appeared if the authors had kept only k=4 instead of k=5 clusters? Even if that was not the case, I understand that it might be appropriate to retain this cluster if the characteristics of this region are very different from those in the surroundings (e.g. elevated emissions and confinement of pollution within a basin with little ventilation, distinct annual cycle as seen from Figure 5).

   During our exploratory analysis, in the case of k=4 clusters the sites in the Po Valley appeared together with stations in Southern France and Central and East Spain. We believe that in order to be able to study the special circumstances in this area, it is important to retain these stations in a separate cluster. We must note, that the Po Valley cluster due its small number of sites, is of course not considered a general case, but, findings in Po Valley apply only for these specific sites and area.

   (b) around line 29 of page 9: "The sites in "PoValley" display the most pronounced S(t), mainly due to the Mediterranean weather conditions, e.g. high temperatures. At the same time high NOx and VOC emissions in this region leads to higher O3 concentrations". I am not convinced at all with this statement. Note that the amplitude of the S(t) component is remarkably wider both for the Po Valley and the Central North cluster compared to the others

(Figure 4). I am surprised at the results for the Central North cluster, where I would expect average ozone concentrations during the warm season (but not in the colder months) to be clearly below those in the Po valley. The authors should explicitly mention this similarity between two apparently very different regions and, if possible, explain why this happens. In other words, are there any reasons why the impact of meteorology and emissions on ozone presents stronger seasonality in these two regions than in others? In addition, I would remove "Mediterranean weather conditions, e.g. high temperatures", which I find quite vague. I think the expression "Mediterranean weather conditions" is much more appropriate for the coastal sites in Spain, southeastern France and in the proximity of Rome (see Figure 1). I am not sure that "e.g. high temperatures" is appropriate either here because this analysis includes ozone data in all seasons.

The Central North high seasonal values are probably related to the industrial and agricultural emissions, and is now mentioned in the manuscript as follows: "It is interesting to note that both Central North sites have seasonal values in their signal comparable to the Po Valley values, probably related to industrial and agricultural emissions in the area of Northern Germany." Higher temperatures in Po Valley compared to rest Central and Northern Europe in combination with the high emissions and topographic conditions (valley south of Alps) that trap the emitted pollutants and retain cyclonic systems in the area lead to the observed high concentrations. To make this more clear we added the following statement: "Special topographic conditions (valley south of Alps) contain emissions from the Milan industrial area in combination with cyclonic systems (????)."

3. In the long paragraph between lines 4-21 of page 10 the authors compare the results to those of previous classifications, namely Carro-Calvo et al. (2017) and Lyapina et al. (2016). See comments:

(a) The comparison of the results of the S(t) clustering to those by Carro-Calvo is probably too exhaustive. I would simplify it, but this is up to the authors to decide whether they want to do that. Rather than mentioning every single regional difference arising from the comparison of both classifications, I would instead list all the possible reasons why the results of both classifications are expected to differ. Only some of those reasons are mentioned in the text. Basically, Carro-Calvo used a MDA8 O3 gridded dataset considering only the summer months, while daily O3 at specific sites during the whole year is used here. In addition, Carro-Calvo applied k-means on normalized anomalies while the spatial classification presented here is based on the seasonal component. Finally, the authors are right to indicate that some of the clusters of Carro-Calvo et al. (2017) do not appear here because the former study used gridded data over locations with few observations, but this explanation is not complete. Note that the final number of clusters will depend on the a-priori choices made (e.g. decisions on the number of clusters based on the explained variance achieved, intra-cluster variance or RMSE, minimizing correlations among different clusters, silhouette width, and so on).

Indeed, the comparison would suffice by explaining the reasons of the differences between both studies. The respective part of the manuscript is now as follows: "Compared to **?**, similar geographical clusters were identified here, except for the Iberian Peninsula, eastern Europe, northern Scandinavia and the Balkan states that do not appear as separate clusters in our analysis. This is most probably due to the small number of observational sites in the above regions. In contrast to our study, gridded MDA8 $O_3$ concentration during summer have exclusively been used for the cluster analysis by **?**, therefore conditions when the correlation of $O_3$ and meteorological variables such as temperature is typically strongest. In addition, the present study results in spatial classification by utilizing the seasonal variation, while **?** have used normalized anomalies." It is true that number of clusters depend on a-priori

choices. We believe that in addition to the above argument, the reason for this difference here is that our data set completely lacks data points in the regions of eastern Europe, northern Scandinavia and the Balkan states.

(b) I feel the comparison of the results from the L(t) clustering to those of Lyapina et al. (2016) would benefit from some additional explanations. That work performed two cluster analyses (CA). The first CA used absolute mixing ratio values and resulted in 5 clusters (Table 2 of that paper), while the second CA used normalized mixing ratios and yielded 4 clusters (Table 3 of that paper). As it is not straightforward to summarize the description of the clusters in those tables, one could just select one of them (e.g. the first one) and provide some simple explanations. For instance, one could indicate that the results from this study are similar to those of a classification by Lyapina et al. (2016) who found 5 clusters of different type, ranging from urban traffic (equivalent to the "highly polluted" reported here) to rural background.

Indeed, it is useful to strengthen this comparison. We changed this part to the following sentence: "Four site type clusters were found based on the LT(t) in this study similar to **?** based on absolute mixing ratios of $O_3$ variations, which identified five site type clusters ranging from urban traffic (as the "HIG" cluster here) to rural background environments (equivalent to "RUR"). "

(c) It is very good that the authors have acknowledged previous work and compared their results to those studies. Apart from that, either here or somewhere else in the paper, I would emphasize the strength of this work: they authors have clearly gone a step forward compared to those studies because they have provided a two-dimensional classification.

We have updated part of the Conclusions section with two additional sentences. "Such a two-dimensional site classification scheme provides an powerful approach for $O_3$ trends studies in large spatial domains and can be of significant use in model evaluation studies (e.g. **?**)."

4. Figure 5 (Annual cycle of daily mean O3 S(t) for the daily mean S(t) clusters) appears on page 12, but I think it is not referenced to from the main text. The figure should be moved to another part of the text (Section 4.4. Ozone seasonal cycles), which would affect the numbering of other figures. Then, in section 4.4, it would be good to mention some of the main features seen from the S(t) component of daily O3 in that figure. For instance, the figure nicely shows that the ozone maxima occurs in summer for the Po Valley cluster and much earlier in the year in the North cluster. This is consistent with previous studies that have reported that both the highest average ozone concentrations and extreme ozone episodes tend to occur over central/southern Europe during summer and over northern Europe in spring (see e.g. Figs 1 of both Schnell et al., 2015 and Ordonez et al., 2017). Finally, I would explicitly mention the days of the ozone maxima in each cluster when commenting the trend of DoMax in Table 2.

This is right, the figure with the annual cycles fits better to section 4.4 and shows more explicitly the differences in annual cycles across the different parts of Europe. We added the following sentences in the manuscript: "In Fig. 8 the average annual $O_3$ cycles are shown; it is clear that in Po Valley the day of maximum $O_3$ occurs in summer (June-July), while in the North occurs around spring time (late March-April). This agrees with previous studies, where both the highest average $O_3$ concentrations and extreme $O_3$ episodes tend to occur over the central and southern parts of Europe during summer while over northern Europe during spring (**??**)."

5. Section 4.2 is on trends of daily mean ozone, but Figure 7 at the end of that section shows results of MDA8 LT for Mace Head. Why do you use MDA8 instead of daily O3 for that particular figure? Is it just to compare the results with those of Derwent's papers (see first paragraph of page 14)?

Indeed, the reason for the choice of the MDA8 in this case is a direct comparison to study by Derwent et al (2013). We now clarify this as follows: " .. we estimated

the LT(t) variation of MDA8 O$_3$ and the Theil-Sen trend for the site in Mace Head (Fig.7) to compare with the MDA8 O$_3$ trend identified by **?**"

6. As seen from the first paragraph of section 4.3 (trends of peak O3) the main result from that section is that, unlike previous studies like that of Flemming et al. (2018), the meteorological adjustment results in significantly negative trends at many sites. That is a very nice result, but I am not fully convinced with all the interpretations of the trends in the following paragraph. For instance, around lines 17-20 of page 14: "in the "BAC" cluster (especially the "West" cluster) the decrease of MTDM was not so pronounced, likely due to the increase of hemispheric transport of O3 in Europe (Derwent et al., 2007; Vingarzan, 2004)". However, those papers roughly cover the first halve of the period of analysis, where I agree that might have been the case (see e.g. Figure 7 for a different metric at Mace Head). Moreover, a few lines below (lines 1-3 of page 15) they claim that there might be some connection between the industrialization of Eastern Europe and the trends in some clusters (lines 1-3 of page 15). I admit that these interpretations are plausible and that the authors have been reasonably careful with their statements, but I would add a short sentence to mention that some more dedicated analyses (e.g. modelling studies) would be needed to investigate the reasons for such trends. I fully understand that such analyses are out of the scope of this paper.

The referee is correct, in order to know exactly the source of such behavior, modeling studies are needed. To make it clear we added in the manuscript the following statement: "Nevertheless, in order to estimate the reasons and quantify the exact influence of the above factors on the trends, dedicated modelling studies are needed."

7. I also like the idea of examining the seasonal cycles of O3 in Section 4.4 and the results presented there are relevant. However, I am not convinced about some of the explanations given there as there are some inaccuracies. In addition, I am

not happy at all with the writing and believe that this section has been written in a rush. There are so many inaccuracies and corrections to make (some of them included in the technical corrections section) that it very hard to focus on the science. Examples: * Lines 17-19 of page 16: "The increase in the Smin(t) may be partially due to the . . .and probably due to the increased influx of O3 towards north and northwest Europe and more cyclonic activity in the North Atlantic during winter as well (Pausata et al., 2012)". Apart from improving the writing (too many "ands" within the same sentence), I am not convinced at the explanations regarding Pausata's paper. What you do mean by increased influx and cyclonic activity? Are such things really mentioned that way in that paper? If so please explain this better. As far as I remember, that work simply suggested that the increasing baseline ozone in western and northern Europe during the 1990s could be associated with a prevailing NAO+ phase during that period.

In winter, positive NAO conditions are linked to enhanced westerly flow as well as intercontinental transport of air masses. Thus, the increase of winter O3 values (Smin(t)) might be linked to the increase of baseline O3 that **?** have found in their study, which is related to the positive NAOI. We rephrased our statement as follows and improved the readability as follows: "An increase in baseline $O_3$ related to the prevailing positive NAO Index -and the associated westerly flow and intercontinental transport- during 1990s and beginning of 2000s is probably a factor contributing to the increase of the winter Smin(t) O3 values."

* Lines 24-27 of page 16: "The observed shift of the day of seasonal maximum might be linked to the increase of emissions in East Asia that have contributed to increased transport of air pollution to middle-and northern latitudes (Zhang et al., 2016) where the effect on O3 is probably greater due to greater convection, reaction rates and NOx sensitivity [some refs.]. . .". Need to completely rewrite this sentence because it is hard to understand. I assume that the strong convection takes place in East/Southeast Asia instead of at mid/north latitudes as it

reads now from this sentence. In addition, the word "greater" is repeated within the same line.

We have rewritten this sentence as follows: "The observed shift of the day of seasonal maximum might be linked to the increase of emissions in East Asia. The associated strong photochemical reaction rates, convection and $NO_x$ sensitivity in the tropics and subtropics (**????**) have probably contributed to increased transport of air pollution to middle and northern latitudes (**?**)."

\* Line 31-33: "The positive phase of the NAO leads to increased O3 concentrations in Europe through higher westerly winds across the North Atlantic, and enhanced transport of air pollutants from North America to Europe (Creilson et al., 2003)." All this looks a bit redundant. Do you simply mean that "The positive phase of the NAO leads to increased O3 concentrations in Europe through enhanced transport of ozone and precursors across the North Atlantic from North America to Europe (Creilson et al.,2003)"?

That is right, we updated this sentence: "The positive phase of the NAO leads to increased $O_3$ concentrations in Europe through enhanced transport of $O_3$ and precursors across the North Atlantic from North America to Europe **?**."

**Technical comments and corrections**

- Line 3 of page 1 (abstract): "the effect of these reductions on ozone is investigated by analyzing surface measurements of ozone". Change the second "ozone" to "this pollutant" just to avoid redundancies.

  Done.

- Lines 13-14 of page 1: "The effect of hemispheric transport of ozone can be seen either in regions affected by synoptic patterns in the northern Atlantic or at sites located at remote high altitude locations". I do not consider this as an

appropriate sentence for the abstract. The manuscript includes some references on the impact of long-range transport and changing weather patterns (e.g. impact of the NAO), but it does not provide any supportive evidence of the relevance of such processes.

That is true, we excluded this sentence from the abstract.

- Lines 17-18 of page 1: "while seasonal cycle trends and changes in the sensitivity of ozone to temperature are driven by regional climatic conditions". I would tone down this statement. Honestly, I do not think that the manuscript proves that this impact is larger than that of the varying rates of ozone precursor emission reductions over the different regions.

To make it more clear that the climatic factors are indeed influencing the trends but are not the only factors that play a role in the observed trends we changed this statement to the following: "while seasonal cycle trends and changes in the sensitivity of $O_3$ to temperature are among other factors driven by regional climatic conditions."

- Last line of page 2 (Introduction): "For instance, in the U.S., O3 climate penalty – defined as the slope of the O3 versus temperature relationship – dropped from 3.2 ppbv/C before 2002 to 2.2 ppbv/C after 2002 as a result of NOx emission reductions (Bloomer et al., 2009)". One could add a reference to Colette et al. (2015), who analysed chemistry-transport and climate-chemistry model projections to asses the impact of climate change on this climate penalty over Europe by the turn of the century.

This is right, we added the following sentence: "Additionally, Colette et al (2015), based on chemistry-transport and climate-chemistry model projections, assessed the impact of climate change on the climate penalty and found that over European land surfaces summer $O_3$ change is [0.44; 0.64] and [0.99; 1.50]ppbv

(95% confidence interval) for the 2041-2070 and 2071-2100 time periods, re-
spectively."

- First line of page 4: "during May and September" -> between May and September

  Done.

- 5th line of page 4: Did you use surface pressure or sea level pressure (SLP)?

  We used surface pressure.

- Line 21 of page 5: Move "e.g." to the beginning in "(Lyapina et al., 2016, e.g. )".

  Done.

- Line 17 of page 6: where yd (t) the de-seasonalized -> where yd (t) is the de-
  seasonalized

  Done.

- Line 19 of page 6: "of the" can be removed from "de-seasonalized concentrations
  of the yd (t)".

  Done.

- Line 25 of page 6: "because" is repeated within the same sentence.

  The sentence is now improved to the following: "A different approach for meteoro-
  logical adjustment was used for the peak $O_3$ than for the daily mean and MDA8;
  de-seasonalization is not meaningful for peak $O_3$ because peak $O_3$ events are
  temporally localized."

- Lines 4-5 of page 7: "For the GAMs, the following meteorological variables were
  used". I would remove "meteorological". Reason: the Julian day, which is not a
  meteorological variable, is also included in the model.

We corrected this, by specifying that the Julian day is a time variable. "For the GAMs, the following meteorological variables were used: the daily maximum temperature, daily mean specific humidity, daily mean surface pressure, daily maximum boundary layer height, morning mean convective available potential energy (CAPE), daily mean East-West surface stress and daily mean North-South surface stress, as well as a time variable the Julian day."

- Line 6 of page 7: Again "daily mean surface pressure". Do you mean SLP?

  Surface pressure was used in the models.

- Line 7 of page 7: No need to spell out CAPE again.

  Right, this has been corrected.

- Line 8 of page 9: "mostly located at higher altitudes". Higher than what? "Higher" could be changed to something like "relatively high" or "elevated".

  We changed to "relatively high altitudes".

- Lines 9-10 of page 11: "the positive trends can be partly explained by . . . originating from the diesel vehicles". Change to "the positive trends could partly be explained by . . . originating from the proliferation of diesel vehicles"

  Done.

- Line 6 of page 12: "rural sites and small and non-significant" -> "rural sites as well as small and non-significant"

  Done.

- Figure 6: Please indicate what the box-plots indicate (i.e. median, edges of the boxes: 25-27th percentiles, whiskers related to interquartile range or to some specific percentiles, etc.).

[Figure]

We clarify this by adding the following sentence. "Boxes include 25th to 75th percentiles with the line indicating the median value; whiskers extend to 1.5 times the interquartile range."

- Lines 7-8 of page 13: "Trends were estimated 0.08 ppb/year [0.06,0.1] for the first period and -0.04 ppb/year [-0.09,0.02] for the second period". Lines 1-2 of page 14: "Derwent et al. (2018) have found an increase of 0.34 $\pm$ 0.07 ppb/year with a deceleration rate after 2007 of -0.0225$\pm$ 0.008 ppb/year". If possible, indicate if the uncertainty estimates correspond to the 95% confidence intervals or to something else. There are other parts of the text where this is not clear.

  In the Methods Section 3.3 it is mentioned that the 95% confidence interval is used for the calculation of the trend.

- Section 4.2 is on the trends of daily mean ozone, but Figure 7 at the end of that section shows results of MDA8 LT for Mace Head. The results look convincing, but why do the authors use MDA8 instead of daily O3 for that particular figure? Is this just to compare the results with those of Derwent's papers (see first paragraph of page 14)?

  Yes, the MDA8 is used for direct comparison to the result by Derwent et al (2018).

- Lines 6-7 of page 14: The word "increased" can be removed from both lines.

  Done.

- Lines 1 and 5 of page 16: "central Northeast Germany" and "central Northeast region". Do you mean "central and northeast"?

  Indeed, the right expression is "central and northeast".

- Line 28 of page 16: "meteorological factors have affected" -> "meteorological factors may have affected".

Done.

- Line 30 of page 16: Add space before while in "data,while".

  Done.

- Line 34 of page 16: "resulted to" -> "resulted in" (this should be changed somewhere in the supplement too).

  Done.

- Caption of Figure 9 on page 17: Change "pm" to "+/-" in "average pm the standard deviation".

  Done.

- Line 2 of page 17: "more increased" -> "increased"

  Done.

- Lines between pages 17 and 18: "The early spring maximum in the "North" sites in April can be explained by higher NOx that is released from PAN and . . .". What do you mean by "higher NOx"? Higher than what? Do you mean something like elevated NOx? By the way, why don't you refer to Figure 5 here (see comment above)?

  "Elevated NOx" is indeed a more appropriate term. Figure 5 is now moved to this section as Figure 8 and added as a reference in this sentence.

- Table 2 on page 18: According to the methods section, one should write "SDoM". Need to change that in the caption and header of last column.

  Done.

- Lines 6-9 of page 19: "At PoValley sites the decrease was more pronounced (-0.083 ppb/K/year). At the same time the average correlation between O3 and

temperature is the highest compared to the other regions, because of large reductions of precursors concentrations in this region which is characterized by high industrial emissions". I assume you mean something like "At PoValley sites the decrease was more pronounced (-0.083 ppb/K/year) because of large reductions of precursor concentrations in this region which is characterized by high industrial emissions. Note that the average correlation between O3 and temperature in that cluster is the highest compared to the other regions".

Yes, that is right. We changed the sentence to the above suggestion.

- Lines 9-11 of page 19: Regarding the low correlations between O3 and T in the North cluster I would also mention the low temperature ranges observed there compared to the other clusters (Fig 11).

We already discuss this feature in the following sentence: "This is expected because at these high latitudes mean temperature is lower compared to other regions in Europe (Fig. 11), thus, photochemical production of $O_3$ is weak during the time when $O_3$ typically reaches its maximum concentration." Nevertheless, we added in the text the reference to figure 11 to highlight this difference.

- Lines 12-16 of page 19. Discussion of the stronger trends of the O3-T relationship for the more polluted LT(t) clusters: "Our results are in line with a box-model study that tested the O3-temperature relationship under different NOx level scenarios (Coates et al., 2016). Coates et al. (2016) have shown that at high NOx conditions O3 increases more strongly with temperature, while the increase is less pronounced when moving to lower NOx conditions". As mentioned for the Po Valley S(t) cluster (see a couple of comments above), I don't see a clear relationship. These references are related to the strength of the O3-T relationship but not to the trend of that relationship. I assume that you mean that regional ozone production has mainly decreased at the most polluted locations, due to considerable reductions of precursor emissions there. Need to rewrite.

Yes that is a valid argument. We have made it more clear to the reader by adding the following sentence according to the referee's comment: "Consequently, regional $O_3$ production has mainly decreased at the most polluted locations, due to considerable reductions of precursor emissions."

- One can remove the columns "standard deviation" in Tables 3 and 4. The p-values should suffice.

That is true, but we believe that it gives a good perspective on the range of variation in the trend magnitudes.

- In different parts of the text, the authors indicate that the S(t) clusters represent the "climatic conditions". I would add the word "regional" to clearly indicate that this is indeed a geographical classification that clearly reflects the regional climate conditions. This can be done in different parts of the text. Here I simply include an example for lines 5-6 of page 21: "Our approach captures several features of O3 variations, i.e. pollution level from the L(t)-clustering and influence of the climatic conditions from the S(t)-clustering". I would change the end of this sentence to "influence of the regional climate conditions from the S(t)-clustering".

Indeed, "regional climate conditions" is a more accurate expression. We changed this in several parts of the manuscript.

- Lines 17-18 of page 21 (Conclusions): "peak O3 has been decreasing with the smallest rate at higher altitude sites especially in the western part of Europe due to the influence of background O3 imported from North America and East Asia". Are the evidences for this long-range-transport influence so clear? If not I would add the word "possibly" before "due to".

That is right, the argument was not proven here. We added the word "possibly" in the statement.

- Line 24 of page 21: "the sensitivity of O3 to temperature has weakened since 2000 with a rate of around 0.084 ppb/K/year". It should be considerably less than that because that value is only found for the 4 sites in the Po Valley cluster (see Tables 3 and 4).

  True, this is a typing error. The true average value is 0.04 ppb/K/year.

- Lines 25-26 of page 21 (about the decreasing O3-T slope): "It was shown that differences in changes to this sensitivity across sites are mainly driven by regional meteorological conditions". I do not see any proof of this in the manuscript. It might well be related to varying rates of reductions of precursor emission across the different regions. I have a similar comment about this in the abstract so both of them can be addressed at the same time.

  The main argument here is that the trend differs amongst the regional clusters, thus, the climatic conditions probably influence this trend. Nevertheless, we rewrote this part to clarify this argument as follows: "Finally, the sensitivity of $O_3$ to temperature has weakened since 2000 with a rate of around 0.04 ppb/K/year, i.e. formation of $O_3$ became weaker at high temperature conditions, that can be attributed to the decrease of $NO_x$ concentrations. The trend of the sensitivity differs across sites that are influenced by different meteorological conditions."

- The references Boleti et al. (2018a) and Boleti et al. (2018b) should be changed to Boleti et al. (2018) and Boleti et al. (2019), respectively. The second paper has been published and should be updated in the reference list.

  Done.

- Figures S1 & S2: Can you please enlarge fonts. They are too small and very hard to read.

  Done.

- Lines 2-3 of page 3 in Supplement: "objects have a low similarity with the rest objects" -> "objects have low similarity with the rest of the objects"

  Done.

- Figure S3: Even when there are some explanations on the previous page, most readers are not very familiar with the concept of silhouette width. In the figure caption I would indicate (1) that each horizontal bar represents the silhouette width for a particular site in a given cluster and (2) that this parameter is clearly positive for most sites.

  We added the following in the caption: "The bars indicate the value of the $S_W$ for a particular site within the respective cluster. For the majority of the sites the $S_W$ is positive showing high similarity within the clusters."

- Figure S5: I assume this is for W(t) instead of for S(t).

  That is right, we corrected to W(t).

- Section S3 (Additional information on clusters): What is the use of this section if you hardly provide any comments e.g. about the MDA8 O3 clusters in the main manuscript? As indicated above, the main text and most of the supplement seem detached from each other.

  We believe that it is interesting to show that both metrics lead to similar results. Also, we have now connected this Section to the main manuscript, as described in response 2 of the main comments.

- Lines 2-4 of page 10 in supplement: "In this section we present more detailed information about the clusters extracted from the daily mean and MDA8 O3 LT(t), S(t) and W(t)". Please refer to Figures S9-S11 there.

  Done.

- Caption of Figure S9: Need to remove "Map indicating the sites that belong in each cluster and average LT(t) in each cluster with the standard deviation of the sites that have SW>0".

  Done.

- Captions of Figures S10-S12: Need to use subscript for W in "SW" (silhouette width, defined in section S2).

  Done.

- Lines 4-5 of page 14 in supplement: "The level off or small increase in the High-Poll stations can be attributed to the smaller rate of reduction of VOCs, which resulted to reduced titration of O3 by NO". I agree with the reduced titration due to the decrease of NOx emissions, but do you have any evidence about the smaller rate of reduction of VOCs? If so one should provide a reference. A simpler explanation might be a change of chemical regimes, i.e. in the sensitivity of O3 production to NOx and VOCs.

  The smaller reduction rate of VOCs can be seen in the report by (?). This reference is now added to the text, as well as the suggested argument about the chemical regimes. "An additional explanation for the observed trend might be a change of chemical regimes , i.e. in the sensitivity of $O_3$ production to $NO_x$ and VOCs."

- Line 6 of page 14 in supplement: Change "(Fig. S14" to "(Fig. S14)". Then remove "respectively" because it is not needed there.

  Done.

- Lines 10-11 of page 14 in supplement: "Here, the sites with negative SW that were not considered in the discussion of the trends are presented. In the LT(t)-clustering four sites with negative SW were identified (Fig. S15), in the S(t) 26

sites (Fig. S16) and in the W(t) 24 sites". It is unclear whether you are talking about the clusters of daily O3 or MDA8 O3.

It is the daily mean $O_3$ clusters, we have corrected this to "daily mean $O_3$ LT(t)-clustering".

• Caption of Figure S15 is not complete. Need to indicate the meaning of the different lines/shading.

Done.

• Caption of Figure S16 is not complete either. For instance you don't show the "clusters average S(t)" as you claim because there is some spread in the figure. Probably mean +/- standard deviation?

Right, it is the average $\pm$ the standard deviation. The caption is now improved as follows. "Example cases of sites with negative SW in the S(t)-clustering (black dashed line) in comparison with the clusters average S(t) $\pm$ the standard deviation (shaded area)."

---

## Author Comment (AC2) · 19 Feb 2020

**Response to reviewer comments: "Temporal and spatial analysis of ozone concentrations in Europe based on time scale decomposition and a multi-clustering approach"**

Eirini Boleti, Christoph Hueglin, Stuart Grange, Andre Prevot, Satoshi Takahama

**Response to reviewer 2**

The manuscript presented by Boleti et al. examines trends on surface ozone concentrations across a number of stations over Europe for the period 2000-2015. They use a time-scale decomposition to analyse long-term (LT), seasonal (S) and short-term (W) variations. Then, they apply a clustering technique and they finally calculate the trends in the clusters obtained. In addition, they analyse the ozone-temperature relationship over the different clusters and sub-periods. Their classification is consistent with previous studies and their results show a general decreasing in the ozone concentrations, mostly in the ozone peaks. In addition they find a reduced sensitivity in the ozone- temperature relationship over most of the clusters defined. Overall, I found the manuscript very interesting and complete. The methodology applied is robust and consistent, as well as the results presented. However, I also think that there are some parts in the current version that should be improved in order to be published, in particular the methods sections (see my comments below). In my opinion the manuscript might be a good

contribution to Atmos. Chem. Phys and the scientific community. Therefore, I would be happy to support the publication of the present manuscript after addressing some comments, which I consider that would be useful to improve it. I have a few general comments and some specific comments:

**Main Comments**

There are some parts in the methods section that are not very clear, and in my opinion this section is essential to follow the manuscript. Therefore I have some comments and questions that I would like to ask the authors:

1. Section 3.1. Time scale decomposition (page5): The authors should explain in more detail the IMF. How the number of coefficients ($c_j$) is selected? The authors say "By adding together IMFs with frequencies around 40 days and 3 years we obtain the seasonal variation of O3 ($c_1$ +...$c_{10}$)", but why 40 days and 3 years? Is this based on the previous study from Boleti et al.2018? I think that this information should be included in order to help the reader to better understand the methodology.

   The selection is discussed in Boleti et al (2018) and for the reader that is more interested on that matter we refer to the above publication. We have added the sentence: "A more detailed discussion on the choice of the IMFs for the seasonal and short term variations can be found in Boleti et al (2018)."

2. Sections 3.3-3.4 Long-term trends (page 6, 7): I understand that for the peaks of O3 metrics the method explain in section 3.3 cannot be applied. But, it would possible to use the same method, i.e. GAMs models also for daily mean and MDA8 O3, wouldn't it?

   Yes, that would be another good approach to estimate meteorological influence of daily mean and MDA8 $O_3$.
3. Why the authors define the warm season as May-September? Why April is not included? I think this should be further clarified, since usually ozone season ranges from April to September (e.g. EEA, 2019, Fleming et al. 2018)

This is based on the estimation of the MTDM which In previous reports by the EEA refers to the period between May and September. Thus, for consistency and to compare our study with the EEA studies, we keep this period of the year as representative for occurrences of peak $O_3$. We now mention the above in the manuscript: "The models are fitted for the warm season May-September as by definition the MTDM refers to this period of the year."

4. Reading the modelling part (section 3.3, 3.4, page 6) is not clear the input data to calculate the trends, e.g. the GAM models are fitted to each cluster that contains a number of stations, so the models are applied individually to each station, am I right?

A GAM model is applied to each station separately and the trends are calculated for the individual stations as well.

5. Regarding the analysis of seasonal cycle of O3, why do the authors chose the mean of O3 and not the MDA8 O3?

The reason we use daily mean O3 in our analysis is to be consistent throughout the manuscript and be able to make comparisons between the different O3 metrics and clusters.

**Specific comments**

1. L26-30 of page 3. The authors applied a filter to obtain the time series, and only those with a maximum of 15% of missing values and maximum of 120 consecutive days are used. Is this 15% applied to whole period (16 years) or each year?

And the consecutive days? I assume that they refer those 120 consecutive days in one year, is that correct? Can the authors clarify this?

15% missing data and the 120 consecutive days refer to the whole period of measurements. Thus, we clarify this with the following addition: "time series with a maximum of 15% of missing values, and a maximum of 120 consecutive days with missing values are used for the whole period of measurements, leaving the study with 291 sites across the European domain."

2. L1 of page 5. I would add that the clusters are identified by using each component L(t), S(t), W(t) separately to the algorithm.

The following sentence is already mentioned in the manuscript: "For identification of the clusters the LT(t), S(t) and W(t) of the daily mean and MDA8 $O_3$ were used as input time series in the PAM algorithm."

3. L4 of page 8. What are the temperature ranges considered?

The temperature ranges from 7 to 35 degrees Celsius in intervals of four degrees; (7,11] (11,15] (15,19] (19,23] (23,27] (27,31] (31,35]

4. L4 of page 9. Why do the authors leave the results of MDA8O3 in the supplement and the results if the O3 in the main text? Wouldn't it be more interesting to see the results for MDA8O3?

We refer to our answer of the main comment 1 from referee 1. In summary, O3 daily mean and MDA8 clusters are rather similar, the choice of clusters between the two metrics does not affect the conclusions. In addition, for comparison with other studies and consistency throughout the manuscript, we believe it is more interesting to present the daily mean O3 clusters.

5. L19-23 of page 9. Please refer to figure 3.

Done.

6. L25 of page9. Just a comment regarding Fig.4. the colours for "Po Valley" and "Central South" maybe could be changed, they are quite similar and it is hardly to distinguish the stations that belong to each cluster.

This is a valid remark, we changed the color of the Po Valley cluster to a darker blue color.

7. L4-L20 of page 10. Should Figure 5 be referred here? I couldn't find any reference to Fig.5.

This is true, the figure is now moved to the section 4.4. O3 seasonal cycle trends, after a similar suggestion of referee 1.

8. L3 of page 11. "decreasing O3 trends", maybe it should be specified "decreasing daily O3 means".

That is right, it is corrected to "decreasing daily O3 means".

9. L4 of page 11. In my opinion, the table 1 with the number of stations should be introduced before (e.g. when presenting the clusters).

Indeed, we have moved this table in section 4.1. Cluster analysis.

10. L9 of page 14. Where these percentages 62% and 18% came from exactly? Is there any figure to support this? This question is in the line of one my previous comment (i.e. how the models are fitted).

We do not have a figure to support this. These percentages refer to the sites, where negative trends were estimated from all studied sites in our data set, i.e. ratio: sites with negative trends/number of all sites, 62% before and 18% after meteo-adjustment.

11. L10 of page 19. In the North sites the variability of temperature is lower and O3 is also more influenced by transport.

This is indeed a useful addition to the manuscript. "In addition, in these northern regions the variability of temperature is lower compared to the central and southern parts of Europe, while $O_3$ concentrations are more influenced by intercontinental transport mechanisms."

12. L24-26 of page 21. The authors attributed the decreasing O3-Temperature to NOx reductions, and this is likely one of reasons (but it is not showing here) and then, they mention that "changes in the sensitivity across sites are mainly driven by regional meteorological conditions", so what about the NOx emission reductions just mentioned? I think this last paragraph is important and it must be rewritten.

We rewrote this paragraph to the following one: "Finally, the sensitivity of $O_3$ to temperature has weakened since 2000 with a rate of around 0.04 ppb/K/year, i.e. formation of $O_3$ became weaker at high temperature conditions, that can be attributed to the decrease of $NO_x$ concentrations. It was shown that the trend of the sensitivity differs across sites that are influenced by different meteorological conditions."